# CoreCodeBench: A Configurable Multi-Scenario Repository-Level Benchmark

**Lingyue Fu**[1]**, Hao Guan**[1]**, Bolun Zhang**[1]**, Haowei Yuan**[1]**, Yaoming Zhu**[2]**,**
**Jun Xu**[3]**, Zongyu Wang**[3]**, Lin Qiu**[3]**, Xunliang Cai**[3]**, Xuezhi Cao**[3]**,**
**Weiwen Liu**[1]**, Weinan Zhang**[1]** Yong Yu**[1]

[1]Shanghai Jiao Tong University, [2]AGI-EVAL, [3]Meituan

## Abstract

As Large Language Models (LLMs) demonstrate increasingly sophisticated code processing capabilities, evaluating their performance on engineering-level code remains challenging. Existing repository-level benchmarks primarily focus on single scenarios, such as code generation or bug fixing, without adequately capturing the diversity and complexity of real-world software or project engineering workflows. Furthermore, these benchmarks suffer from limited controllability in question positioning and reliability issues in their generated test cases. To address these limitations, we present CorePipe, a fully automated pipeline that converts repositories into comprehensive benchmark test cases, and introduce CORECODEBENCH, a configurable multi-scenario repository-level benchmark. To simulate real engineering scenarios, CorePipe generates three types of atomic questions (Development, BugFix, and Test-Driven Development) specifically targeting core code segments. These atomic questions are further combined into three types of composite questions, with difficulty levels flexibly adjusted through hyperparameter tuning. CORECODEBENCH provides a comprehensive and extensive repository-level benchmark to investigate the applicability of LLMs in real-world engineering projects. Experiments with 16 LLMs across diverse scenarios reveal varying capabilities and offer multi-dimensional insights into LLM performance in engineering contexts. Code of CorePipe and data of CORECODEBENCH are available.

## 1 Introduction

With the continuous improvement in the code processing capabilities of Large Language Models (LLMs), more researchers are starting to focus on their applications in engineering-level code. Engineering-level code often involves complex dependencies and long-context interactions, posing unique challenges for LLMs. Specialized code LLMs, such as QwenCoder [16] and DeepSeek-Coder [13], have demonstrated exceptional programming capabilities in software engineering. LLM-based products such as Copilot, Windsurf, and Cursor, significantly reduce the complexity programmers face in engineering-level projects. As the code processing capabilities of LLMs continue to evolve, there is a growing need to systematically understand their strengths and limitations across different engineering scenarios. To assess the programming capabilities of these tools at an engineering level, it is crucial to establish an effective and fair evaluation standard.

Several benchmarks have been proposed, such as SWEBench [17], REPOEXEC [14], and Big-CodeBench [38], to evaluate the ability of LLMs in implementing engineering-level code. These benchmarks are derived and refined from real-world repositories, ensuring a high degree of alignment with real engineering code development. They focus on tasks such as natural language to code

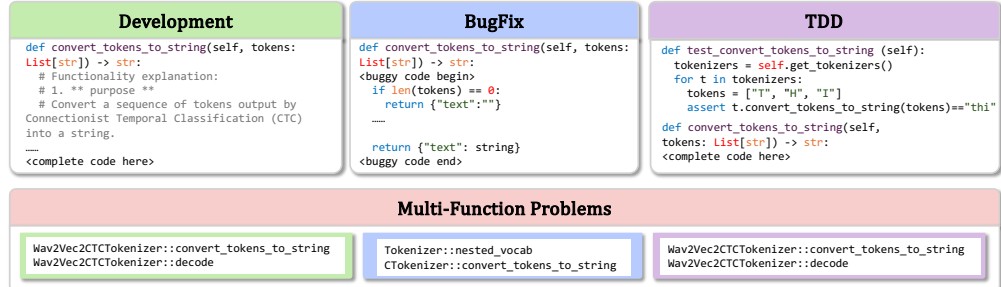

Figure 1: **Overview of CORECODEBENCH.**

(NL2Code) translation and bug fix within the scope of engineering code development. Although existing benchmarks provide an initial reference for evaluating the programming capabilities of LLMs in engineering environments, the current evaluation framework faces two critical challenges.

***Challenge 1: Single Scenario.*** Prevailing repository-level benchmarks primarily focus on the code generation task, and do not adequately encompass the diverse scenarios present in engineering development. In real-world engineering practice, developers not only need to complete function-level code completion but also engage in bug fixies for unit tests. Additionally, within modular development paradigms, engineers often need to simultaneously implement main functions alongside supporting utility functions. These scenarios require the LLMs to display not only code generation capabilities but also cross-file contextual reasoning and implementation planning abilities—skills that current evaluation systems fail to systematically assess.

***Challenge 2: Lack of Controllability and Reliability.*** Existing automated generation methods exhibit significant shortcomings in both controlling the positioning of generated questions and ensuring their reliability, directly impacting benchmark's effectiveness. The random masking approach, while achieving positional randomness, lacks logical constraints in mask selection, which might result in overlooking critical code segments or excessively testing non-essential areas [37]. Alternative approaches such as those based on cleaning pull requests, fix testing locations to historical revision points, limiting evaluation scenario diversity [17, 29]. These methods also suffer from low data reliability, with numerous pull requests not being self-contained and requiring substantial manual cleaning [28]. Neither method effectively ensures flexible control over test positioning while maintaining core code relevance and data quality, hindering comprehensive assessment of LLMs' performance in engineering-level tasks.

To address these limitations, we design a fully automated pipeline CorePipe that converts GitHub repositories into repository-level benchmark test cases. CorePipe generates three types of atomic questions (Development, Bugfix, Test-Driven Development) on core code segments, and further composes multiple composite question types with adjustable difficulty. Quality inspection and analysis show that the generated data are of high quality and reliability. As shown in Figure 1, we release a meticulously **Co**nfigurable **Re**pository-level benchmark, CORECODEBENCH, which effectively evaluates the actual capabilities and adaptability of LLMs in engineering-level code development. Through comprehensive evaluation of general-purpose and code-specific LLMs, we gain insights into the performance and characteristics of these models across diverse repository-level scenarios. CORECODEBENCH not only enables coarse-grained differentiation of LLM code abilities, but also provides fine-grained analysis of their potential. Flexible control of CorePipe over question difficulty enables CORECODEBENCH to offer a promising platform for future LLM evaluation. Our experiments further highlight several areas for improvement in LLMs' performance on engineering-level projects, paving the way for future advancements in model capabilities.

The contributions are summarized as follows:

- We design CorePipe, a fully automated pipeline for generating LLM engineering code capability tests from repository source code without any human intervention. CorePipe can be adapted to any programming language and any repository.

- We release the analysis and quality inspection results of the test data generated by CorePipe. The results demonstrate that CorePipe can produce high-quality and highly flexible test cases.

| Benchmark | Multi-Task | Automatic | Difficulty Level | Flexible Position | Quality Inspection | Avg. Lines |
|---|---|---|---|---|---|---|
| SWEBench [17] | ✗ | ✓ | ✗ | ✗ | ✗ | 38.01 |
| DevBench [19] | ✓ | ✗ | ✗ | ✗ | ✗ | - |
| ExecRepoBench [37] | ✗ | ✓ | ✗ | ✗ | ✗ | 2.42 |
| Codev-Bench [29] | ✗ | ✓ | ✗ | ✗ | ✗ | 43.69 |
| EvoCodeBench [20] | ✗ | ✗ | ✗ | ✗ | ✗ | 14.86 |
| RepoMasterEval [34] | ✗ | ✓ | ✗ | ✗ | ✗ | - |
| BigCodeBench [38] | ✗ | ✗ | ✗ | ✗ | ✗ | 13.55 |
| REPOEXEC [14] | ✗ | ✓ | ✗ | ✗ | ✓ | 21.9 |
| CORECODEBENCH | ✓ | ✓ | ✓ | ✓ | ✓ | 34.14 |

Table 1: Comparison between existing repository-level benchmarks and CORECODEBENCH.

- We provide CORECODEBENCH, a repository-level benchmark that includes three atomic tasks and three composite tasks. CORECODEBENCH features various question types and characteristics, offering new insights and analytical perspectives for evaluating LLM coding.
- We present the evaluation results on several state-of-the-art LLMs and conduct multifaceted analyses of their performance on repository-level scenarios.

## 2 Background and Related Work

### 2.1 Large Language Models for Code

General-purpose LLMs have demonstrated remarkable performance not only in natural language processing but also in code-related tasks. In recent years, LLMs tailored for code generation and reasoning have consistently achieved high scores in benchmark tests. On the HumanEval benchmark [7], the closed-source models Claude-3.5-Sonnet [3] and GPT-4o-0513 [24] have reached Pass@1 scores of 92.0% and 91.0%, respectively. Among open-source models, DeepSeek-Coder-V2-Instruct [9] and Qwen2.5-Coder-Instruct [16] have achieved Pass@1 scores of 90.2% and 88.4%. On other algorithmic problem benchmarks like MBPP [6], LLMs have surpassed Pass@1 scores of 85%, showcasing their strong performance in this domain. LLMs have also played a crucial role in engineering tasks, as demonstrated by products like Copilot [12], supporting code writing and debugging in extended context scenarios. To further advance coding LLMs, there is an urgent need for repository-level code benchmarks to evaluate performance in engineering contexts.

### 2.2 Existing Repository-level Benchmarks

Over the years, various benchmarks have been created to evaluate models on code-related tasks. Popular benchmarks focus on evaluating code generation (HumanEval [7], MBPP [6]), debugging (DebugBench [33], QuixBugs [15]), and code translation (CodeTransOcean [36]) capabilities. However, these benchmarks primarily target short code snippets and do not sufficiently address longer code generation or complex software engineering challenges.

Recently, with the enhanced code capabilities of LLMs and the support for larger context windows, several repository-level benchmarks have emerged. As demonstrated in Table 1, these benchmarks can automatically extract or generate test cases from real repositories to evaluate the performance of LLMs on repository-level code tasks. However, due to the random masking [37] or cleaning from pull requests [17, 29], the positioning, difficulty, and quality of the test cases are not consistently controlled. Some benchmarks [20, 38] require manual intervention to generate and validate test cases, thus preventing full automation. Furthermore, aside from DevBench [19], which evaluates LLMs' capabilities in software development through multi-stage tasks, most benchmarks [34, 14] have primarily concentrated on code generation within repository-level projects. Consequently, there is a clear need for a configurable, multi-scenario repository-level benchmark to fully assess the potential of LLMs in more complex software engineering contexts.

## 3 Method

In this section, we introduce the design of the CorePipe, including repository preprocessing, single-function problem generation, and multi-function problem generation. CorePipe is capable of identifying and rewriting core code segments to generate 6 types of problems, simulating various situations

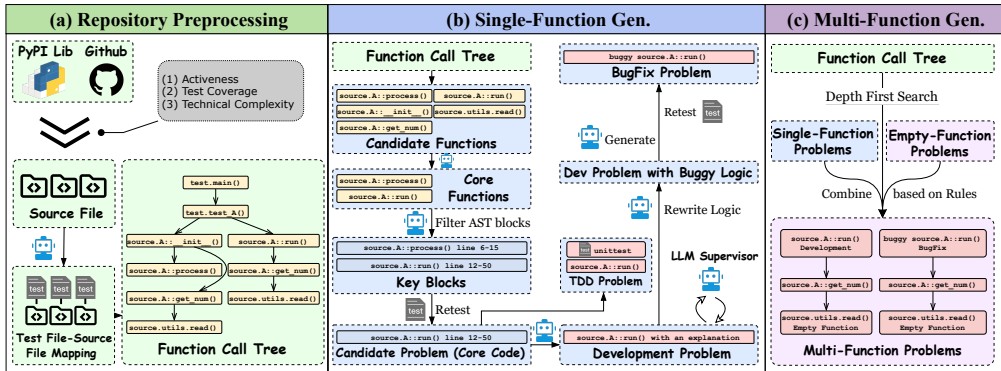

Figure 2: **Overview of CorePipe**. (a) *Repository Preprocessing* selects high-quality repositories based on three criteria, ensuring a diverse and representative codebase collection. (b) *Single-Function Problem Generation* creates three distinct types of problems focusing on individual function understanding and modification, targeting critical code segments. (c) *Multi-Function Problem Generation* constructs complex scenarios requiring an understanding of interactions between multiple functions.

in engineering development scenarios. For both single-function and multi-function problems, our pipeline ensures that the questions are generated from critical and representative locations, maintains the reliability of the generated problems, and allows for controllable difficulty levels.

## 3.1 Repository Preprocessing

**Repository Selection.** The PyPI library is a widely used public repository that offers a vast array of Python packages. We select open-source projects from PyPI based on the following criteria: (1) *Activeness*: the project has been updated or maintained within the past six months; (2) *Test Coverage*: the project contains unit tests, with test files accounting for more than 30% of the codebase; (3) *Technical Complexity*: the project has more than 5,000 lines of code and involves cross-module development. This selection process ensures that the chosen repositories not only reflect real-world engineering practices but also provide a solid testing infrastructure to support subsequent problem generation.

**Test File-Source File Mapping Generation.** We establish the mapping between source files and test files through a process that combines LLM-based analysis and automated rules. Specifically, we (1) use an LLM to analyze the repository's file tree structure; (2) apply automated rules to generate `<source, test>` pairs; and (3) perform executability checks and retain passing tests. The resulting mapping serves as a foundational data structure for subsequent problem generation, ensuring a strong semantic connection between test cases and target source code.

**Function Call Tree Generation.** For each validated test file and source file pair, we perform dynamic tracing on the test file to construct a cross-file function call tree. This process is implemented based on a customized version of the pycallgraph library [18]. Each node in the function call tree represents a function, annotated with its corresponding file and precise location. Every node serves as a potential candidate for Single-Function Problem generation, while the complete function call tree provides the structural foundation for composing multi-function Problems.

Prompts used in repository preprocessing stage is illustrated in Appendix A.

## 3.2 Single-Function Problems Generation

We first generate single-function problems as foundational atomic tasks, encompassing three types: Development, BugFix, and Test-Driven Development (TDD). These atomic tasks are designed to systematically evaluate the abilities of LLMs in long-context comprehension and local code implementation. Throughout the generation process, we dynamically monitor the quality of the questions, ultimately filtering out effective problems that meet the requirements of engineering practice.

**Core Code Identification.** Given that some functions in engineering code are simply basic condition checks or auxiliary utilities without core business logic, we first filter all function nodes in the function call tree to identify *core functions* as problem candidates. For each core function, we automatically select consecutive AST blocks as core code blocks by prompting LLMs to identify key segments, ensuring the completeness and centrality of the extracted segments. The retesting process verifies whether these core code blocks can be effectively detected by unit tests. All core functions and their associated core code blocks that pass the retesting process are considered as candidate problem locations.

**Development Problem.** We mask the identified core code blocks to generate development type problem. We then utilize the GPT-4o [24][1] to generate structured functional descriptions for the masked parts, ensuring that the descriptions cover key information such as input-output specifications, core logic, and boundary conditions. To further enhance the quality of the generated descriptions, we introduce Claude-3.5-Sonnet [4] as a discriminator model to score and provide feedback on the generated paragraphs. If deficiencies are detected, the generation model refines the descriptions based on the feedback. This iterative process is conducted twice. The specific prompt settings for this generation process are detailed in Appendix C.

**BugFix Problem.** Bug fixing is a common scenario faced by developers in real-world engineering projects. For current LLMs, the ability to fix syntactic errors is generally stronger than other error types [21]. Thus we focus more on constructing code snippets that contain logical errors. Specifically, we first use an LLM to rewrite development-oriented problems, generating erroneous logic descriptions for the masked code segments. Then, we employ a smaller-parameter LLM to produce buggy code for these masked segments. In our framework, large models are used to simulate more complex logical errors, while smaller models are used to generate more common and basic errors.

**Test-Driven Development Problem.** Test-Driven Development (TDD) is a software development approach where unit tests are written for target functionality before implementing the actual code. Following the methodology outlined in [22, 1], our TDD problems provide unit tests and require LLMs to implement the corresponding functionality based on these tests. TDD is a promising paradigm for helping ensure that the code generated by LLMs effectively captures the requirements. Specifically, we (1) select unit test code that directly tests specific functions based on the function call tree, (2) mask the core code block, (3) include the unit test code segments in the prompt. With the assistance of the function call tree, we ensure that the source code can be properly reconstructed using contextual information and the unit test.

### 3.3 Multi-Function Problem Generation

In engineering-level software projects, developers often extract parts of an implementation into separate utility functions for reuse. In such cases, a programmer may need to implement several sub-functions while developing a main function. Similarly, during bug fixing, it is sometimes necessary to address bugs across multiple related functions simultaneously. To simulate these real-world scenarios, we design Multi-Function Problems. Each Multi-Function Problem consists of multiple atomic problems, where an atomic problem refers to a single function that needs to be completed or corrected. Atomic problems include four types: development, BugFix, TDD, and empty-function. The Development, BugFix and TDD atomic problems are generated during the single-problem generation stage. For empty-function problems, the contents of utility functions in the repository are removed, leaving only the function signature and declaration. Empty-function problems are used exclusively within multi-function problems.

Each atomic problem corresponds to a node in the function call tree. The combination of atomic problems follows four basic rules: (1) at least one single-function problem is included; (2) the corresponding functions must have a call relationship (i.e., a parent-child relationship in the function call tree); (3) the maximum depth of the call tree is limited to $d$, where $d$ is a hyperparameter; (4) the total number of atomic problems $n$ satisfies $2 \leq n \leq \nu$, with $\nu$ as another hyperparameter. By adjusting the hyperparameters $d$ and $\nu$, we can control the complexity and difficulty of the generated problems. Specific generation rules for different subtypes are provided in Appendix D.

---

[1]Analysis of model selection for data generation is provided in Appendix B.

## 4 CoreCodeBench

### 4.1 Data Statistics

CORECODEBENCH encompasses a diverse col-
lection of 12 repositories covering 6 distinct
repository-level coding tasks, with a total of
1,545 valid problems. Detailed information
about the repositories and illustration of CORE-
CODEBENCH can be found in the Appendix E
and F. In Table 2, we present the key statistics of
CORECODEBENCH, including the average num-
ber of functions, average lines of gold solutions,
and the number of problems for each problem

| Problem Type | # Function | # Lines | # Problem |
|---|---|---|---|
| Development | 1 | 17 | 422 |
| BugFix | 1 | 38 | 433 |
| TDD | 1 | 14 | 276 |
| Multi-Dev. | 3.85 | 53.92 | 167 |
| Multi-BugFix | 2.0 | 62.34 | 10 |
| Mult-TDD | 4.07 | 67.3 | 152 |
| Difficult | 4.75 | 65.66 | 91 |

Table 2: Data Statistics of CORECODEBENCH.

type. The dataset encompasses a diverse range of problem complexities across different categories.

Each problem type contains specific contextual information to facilitate solution generation. Devel-
opment problems include explanations of the masked code segments along with surrounding file
context. BugFix problems contain the buggy code implementation, contextual information, and
optional unit test details to aid in identifying and resolving errors. TDD problems provide file context
and unit test code that defines the expected behavior of the implementation. For Multi-Function
problems, we include code snippets of all relevant functions from the function call tree, offering a
comprehensive view of the interdependent components. Examples of prompts for different problem
types are presented in Appendix G.

### 4.2 Evaluation Metric

We assess the quality of generated code by executing unit tests corresponding to the source code.
Following the method in [7], we adopt Pass@1 as our primary metric. For a given problem, Pass@1
indicates whether the first solution generated by a model successfully passes all associated unit
tests. Additionally, we introduce PassRate as a complementary metric that measures the relative
improvement over the retest baseline. PassRate is calculated as

$$\text{PassRate} = \frac{N_{\text{pass}} - N_{\text{retest}}}{N_{\text{total}} - N_{\text{retest}}},$$

where $N_{\text{pass}}$ represents the number of test cases passed by the solution of model, $N_{\text{retest}}$ is the number
of test cases that pass without any modifications to the code, and $N_{\text{total}}$ is the total number of test cases.
While Pass@1 reflects the ability of a model to generate a fully correct solution in a single attempt,
PassRate provides a finer-grained assessment by measuring the model's incremental improvement
over the baseline, capturing partial correctness across all test cases.

For the overall CORECODEBENCH, both the Pass@1 score and PassRate are calculated as the
average of their respective values across all repositories, providing a comprehensive measure of
model performance across diverse codebases.

### 4.3 Quality Inspection

CorePipe utilizes an LLM supervisor to conduct preliminary quality assessment and filtering of
generated problems. To further ensure problem quality, we implement additional quality inspection
mechanisms specifically for Development-type problems.

**IG Filter.** For LLM-generated explanation texts, we introduce an Information Gain (IG) Score to
measure the informational value provided by the explanations. Specifically,

$$\text{IG}_{\text{base}} = \text{PassRate}_{\text{exp}} - \text{PassRate}_{\text{no-exp}}$$

$\text{IG}_{\text{base}} > 0$ indicates that the explanation provides additional effective information, while $\text{IG}_{\text{base}} \leq 0$
suggests that the explanation information is redundant or incorrect. We select commonly used LLMs
including GPT-4o [24], Claude-3.5-Sonnet [4], Doubao-pro-4k [11], and qwen-plus-latest [2] as
baseline models. Based on the IG scores from these baseline LLMs, we retained only problems with
$\text{IG}_{\text{base}} > 0$ and problems that none of the models could solve (i.e., difficult problems). After applying
the IG filter, 48.56% of the problems are retained.

| Single Function | Development | | BugFix | | TDD | |
|---|---|---|---|---|---|---|
| Models | AC Rate | AC@1 | AC Rate | AC@1 | AC Rate | AC@1 |
| **API** | | | | | | |
| GPT-4o [24] | 82.09 | 57.47 | 57.95 | 34.42 | 84.09 | 46.38 |
| GPT-4.1 [26] | 84.13 | 61.90 | 71.87 | 50.90 | 88.56 | 60.96 |
| o1-mini* [25] | 76.85 | 47.02 | 57.28 | 32.68 | 78.92 | 54.74 |
| o4-mini (high)* [27] | 86.66 | 59.29 | 69.51 | 50.65 | 87.13 | 70.21 |
| Claude-3.5-Sonnet [4] | 86.83 | 61.41 | 63.80 | 40.47 | 85.88 | 60.56 |
| Claude-3.7-Sonnet* [5] [2] | 85.75 | 63.59 | 64.68 | 43.51 | 85.50 | 61.37 |
| Gemini-2.5-Pro-Preview [8] | 73.21 | 48.06 | 30.79 | 22.67 | 74.50 | 51.60 |
| Grok-3* [35] | 80.53 | 56.16 | 54.16 | 33.93 | 84.32 | 53.68 |
| Doubao-pro-4k [11] | 76.25 | 43.54 | 63.19 | 39.43 | 76.10 | 31.24 |
| Doubao-1.5-pro [30] | 84.22 | 57.70 | 64.69 | 41.43 | 83.26 | 45.50 |
| qwen-plus-latest [2] [3] | 78.82 | 52.96 | 39.91 | 22.05 | 80.96 | 40.02 |
| Qwen2.5-max [31] | 83.06 | 57.85 | 50.87 | 28.18 | 82.83 | 47.65 |
| **Open-Source** | | | | | | |
| DeepSeek-Coder-V2-Lite-Instruct-16B [9] | 64.85 | 16.53 | 27.31 | 12.28 | 65.85 | 27.8 |
| DeepSeek-R1* [10] | 84.58 | 58.81 | 66.48 | 45.07 | 79.23 | 56.66 |
| Llama3.1-70B [23] | 71.53 | 41.00 | 51.93 | 28.64 | 79.42 | 37.33 |
| Qwen3-8B [32] | 53.62 | 8.25 | 23.83 | 6.18 | 59.97 | 18.91 |

Table 3: Leaderboard of Single-Function Scenarios. Models using thinking mode are marked with *.

**Manual Inspection.** We further enlist experienced code engineers to annotate the problems. These annotators conducted quality checks on problems that had passed the IG filter. The quality assessment evaluated three aspects: readability, accuracy, and completeness, with flawed test cases being marked as unqualified. We randomly sampled 30 problems from each repository for inspection. Ultimately, the qualification rate for CORECODEBENCH (Development Problems) is 78.55%. This high qualification rate demonstrates that the problems originally generated by CorePipe are inherently reliable. Additionally, we have released the manually verified subset as `CoreCodeBench-Dev-Verified` alongside the main benchmark. We list the detailed experience of three human annotators in Appendix H, where all of them have a bachelor's degree in computer-related major, and at least 3 years of Python development experience.

## 5 Experiments

### 5.1 Setups

**Models.** We present a comprehensive evaluation of a diverse set of LLMs on our proposed CORE-CODEBENCH. The selected models represent a wide spectrum of architectures and parameter sizes, ranging from 7B to 70B parameters. Our evaluation covers both open-source models and proprietary API-based models released by leading AI research organizations. For models that support chain-of-thought (CoT) reasoning, we explicitly enable their reasoning capabilities during inference in order to fully assess their potential for complex reasoning tasks.

**Implementation Details.** All evaluations are performed using the officially recommended inference parameters for each model, including temperature, top_p, and top_k, whenever such recommendations are available. For models without specific recommendations, we employ deterministic sampling settings (temperature= 0, top_k= 1, top_p= 0.0) to ensure reproducible outputs. Other Implementation details specific to other question types are provided in Appendix I.

### 5.2 Main Result of Single-Function Problems

Table 3 presents the performance of various LLMs on the CORECODEBENCH-*Single* benchmark. We draw the following conclusions: (1) *Model Performance*: Claude-3.7 and o4-mini (high) consistently achieve leading results across all three problem types, demonstrating the strong capabilities of recent proprietary models. Among open-source models, DeepSeek-R1 stands out with comparatively better results. Generally, models with larger parameter sizes outperform their smaller counterparts, and newer model versions exhibit clear advancements over previous generations, indicating continuous progress in model architecture and training techniques. (2) *Metric Comparison*: The differing rankings produced by AC Rate and AC@1 indicate that these metrics provide complementary insights into model performance. AC@1 evaluates coarse-grained absolute performance, offering a clear

---
[2]Claude-3.7-Sonnet is a hybrid reasoning model.

[3]In this paper, we use qwen-plus-latest-2025-01-25.

| Multi Function | Development | | BugFix | | TDD | |
|---|---|---|---|---|---|---|
| **Models** | **AC Rate** | **AC@1** | **AC Rate** | **AC@1** | **AC Rate** | **AC@1** |
| **API** | | | | | | |
| GPT-4o [24] | 17.31 | 5.69 | 0.21 | 0 | 18.44 | 6.78 |
| GPT-4.1 [26] | 12.85 | 3.77 | 44.00 | 20.00 | 22.22 | 8.11 |
| o1-mini* [25] | 16.92 | 2.62 | 41.40 | 20.00 | 22.22 | 8.11 |
| o4-mini (high)* [27] | 20.85 | 6.62 | 42.60 | 20.00 | 34.11 | 20.22 |
| Claude-3.5-Sonnet [4] | 24.38 | 7.77 | 41.40 | 20.00 | 24.38 | 7.77 |
| Claude-3.7-Sonnet* [5] | 35.54 | 13.85 | 41.60 | 20.00 | 31.56 | 17.11 |
| Gemini-2.5-Pro-Preview [8] | 22.74 | 6.85 | 2.20 | 0 | 20.22 | 6.89 |
| Grok-3* [35] | 25.62 | 14.46 | 15.40 | 0 | 15.44 | 7.44 |
| Doubao-pro-4k [11] | 3.85 | 0 | 19.80 | 0 | 3.00 | 1.56 |
| Doubao-1.5-pro [30] | 3.08 | 0 | 36.40 | 20.00 | 0.22 | 0 |
| qwen-plus-latest [2] | 21.31 | 8.00 | 27.60 | 0 | 19.22 | 6.89 |
| Qwen2.5-max [31] | 23.46 | 9.31 | 49.20 | 40.00 | 23.89 | 8.22 |
| **Open-Source** | | | | | | |
| DeepSeek-Coder-V2-Lite-Instruct-16B [9] | 0 | 0 | 0 | 0 | 1.22 | 1.22 |
| DeepSeek-R1* [10] | 20.23 | 5.54 | 22.40 | 0 | 23.56 | 9.56 |
| Llama3.1-70B [23] | 19.00 | 4.92 | 37.60 | 20.00 | 19.44 | 6.56 |
| Qwen3-8B [32] | 0 | 0 | 13.8 | 0 | 1.78 | 1.22 |

Table 4: Leaderboard of Multi-Function Scenarios. Models using thinking mode are marked with *.

stratification of code generation capabilities among models. In contrast, AC Rate is able to capture performance differences within the same tier, serving as a finer-grained indicator of a model's potential to pass individual test cases. (3) **Task Comparison**: The relatively lower scores in the BugFix scenario across all models highlight the increased complexity and difficulty of debugging tasks, suggesting valuable directions for future model improvement and research. More detailed results and repository-level breakdowns are provided in Appendix J.

## 5.3 Main Results of Multi-function Problems

Table 4 summarizes the performance of various models on the CORE-CODEBENCH-*Multi* benchmark. Compared to the single-function setting, scores for multi-function problems are significantly lower across all models and scenarios, highlighting the increased complexity and challenges posed by multi-function code generation tasks. Claude-3.7-Sonnet achieves the highest performance among all evaluated models, particularly excelling in the Development and TDD scenarios, which demonstrates its strong generalization and reasoning abilities in more complex contexts.

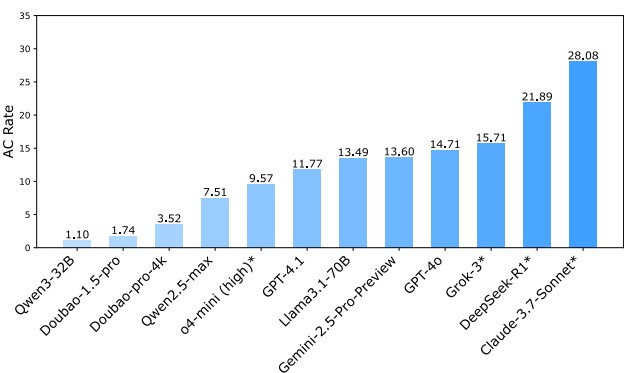

Figure 3: **CORECODEBENCH-*Difficult* Performance**.

Notably, in the BugFix scenario, due to stricter generation rules and a smaller number of available problems, the differences in AC@1 scores among models are less pronounced. However, AC Rate remains effective in distinguishing model performance, as it captures more granular improvements even when absolute success rates are low. More detailed results and repository-level breakdowns are provided in Appendix K

In the multi-function scenario, models are required to provide completions for multiple functions within a single response (see Appendix G for prompt details). Ideally, an LLM would demonstrate planning in its implementation order, such as first completing simple utility functions and then implementing functions that invoke them, or vice versa–reflecting the diverse habits of human engineers.Our analysis reveals that, with the exception of DeepSeek16B-Coder-V2-Lite, most models tend to output answers strictly following the order of the functions as presented in the input prompt. This observation suggests that *current models lack flexible planning and hierarchical reasoning abilities when generating multi-function code*, often defaulting to a sequential approach rather than optimizing for logical or functional dependencies.

CORECODEBENCH-*Difficult* To further guide the development of future LLMs and to push the boundaries of current code generation capabilities, we introduce the CORECODEBENCH-*Difficult* dataset. Specifically, we generate this benchmark by setting the multi-problem generation hyperparameter $\nu = \infty$ (while keeping $d = 3$ to mimic real-world development environments). Figure 3 presents the AC Rate of various models on CORECODEBENCH-*Difficult*. Notably, the pass rates for all models remain below 30%, underscoring the substantial challenges posed by this dataset. These results highlight the effectiveness of the CORECODEBENCH-*Multi* benchmark in revealing the limitations of current models and providing a rigorous testbed for driving future advancements in code understanding and generation.

### 5.4 Coding Capabilities of LLMs

We claim that CORECODEBENCH enables comprehensive evaluation of multiple coding capabilities of LLMs. To visualize these capabilities, in Figure 4, we select nine representative model series and plot radar charts based on their performance across the six distinct scenarios defined in CORECODEBENCH. Each scenario is designed to assess a different aspect of coding ability, thus providing a multi-faceted view of model strengths and weaknesses. For clearer and more intuitive comparison, we normalize the results for each scenario, allowing us to better highlight the differences and relative rankings among models.

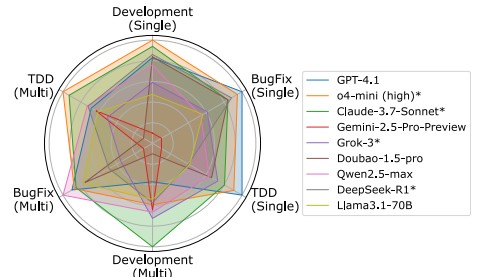

Figure 4: Performance of LLMs on CORE-CODEBENCH across scenarios.

Several key observations can be drawn from the radar charts. (1) The relative ranking of models differs across the six scenarios, indicating that CORE-CODEBENCH effectively evaluates multiple dimensions of LLMs' coding capabilities rather than a single aspect. (2) For Development and TDD problems, model performance in multi-function scenario does not always correlate with that in single-function scenario. This suggests that developing multiple interrelated functions requires additional abilities, such as deeper contextual understanding and implementation order planning. (3) For BugFix problems, model performance in single-function and multi-function scenarios is strongly correlated. This reflects the distinct nature of debugging tasks compared to development tasks, where debugging may rely more on local error correction skills that generalize across different granularities. Overall, these findings demonstrate the value of CORECODEBENCH as a multi-dimensional evaluation framework and highlight the necessity for continued research to develop LLMs with robust and versatile coding skills.

## 6 Conclusions & Limitations

In this paper, we present CorePipe, a fully automated pipeline for generating high-quality, diverse, and controllable repository-level benchmark test cases, and introduce CORECODEBENCH, a configurable benchmark that comprehensively evaluates LLMs' capabilities in real-world engineering scenarios. Through extensive experiments, we demonstrate that CORECODEBENCH enables both coarse and fine grained analysis of LLMs' coding abilities, revealing significant performance differences across various tasks and highlighting areas where current models still fall short, especially in complex and multi-function engineering contexts. Our work provides a scalable and rigorous testbed for the systematic assessment and future improvement of LLMs in engineering-level code development, paving the way for more robust and adaptable AI-driven software engineering tools.

Despite the automated generation of six types of questions from GitHub repositories achieved by CorePipe, our pipeline currently relies on the presence of comprehensive unit tests within the repositories. Repositories lacking sufficient unit tests cannot be processed by our current framework. In future work, we plan to enhance CorePipe by incorporating techniques for generating or augmenting unit tests, thereby expanding its applicability to a broader range of projects. Additionally, CORE-CODEBENCH currently focuses exclusively on Python repositories. We aim to extend support to other major programming languages, such as Java and C++, to enable more comprehensive evaluation of engineering capabilities.

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
