## A  Prompts of Repository Preprocessing

We employ Claude3.5 to analyze the file structure of each repository and automatically identify the main test directories and the source code directories. The prompt used for this purpose is shown below:

```
Below is the file tree of a code repository:
{file_structure}

Please analyze the given file names and paths to identify the corresponding relationships
 between source code and test files (paying special attention to paths containing /test/,
 /unit/, or /unittest/), and provide the output in JSON format. Note that the
correspondence must be based on root path relationships (for example, if both
transformers/test/repo/ and transformers/test/utils/ exist, select transformers/test/).
If specific unit tests exist, the relationship should be detailed to the unit test folder
 (such as unit), and the correspondence can tolerate some missing files as long as the
files generally correspond. If there are no similar corresponding relationships, please
output an empty JSON object.

Example Input:
```
- mlflow/gateway.py
- mlflow/gateway/providers.py
- mlflow/gateway/schemas.py
- mlflow/gemini.py
- mlflow/groq.py
- tests/test_gateway.py
- tests/gateway/test_providers.py
- tests/gateway/test_schemas.py
- mlflow/core/pipeline.py
- mlflow/core/pipeline/graph.py
- core_tests/pipeline.py
- core_tests/pipeline/graph.py
```
Example Output:
```
{
    "repo_name": "mlflow",
    "testcase_dir_mapping":{
        "mlflow/": "tests/",
        "mlflow/core/": "core_tests/",
    },
}
```

Note that after obtaining the mapping, perform a check to merge paths for repeated
occurrences of upper-level directories; remove paths for non-core code segments (such as
cli, community, _sdk, _cli/, etc.); and merge paths in cases where possible. For example:
```
{
    "repo_name": "langchain",
    "testcase_dir_mapping": {
        "libs/cli/langchain_cli/": "libs/cli/tests/unit_tests/",
        "libs/community/langchain_community/": "libs/community/tests/unit_tests/",
        "libs/core/langchain_core/": "libs/core/tests/unit_tests/",
        "libs/langchain/langchain/": "libs/langchain/tests/unit_tests/",
        "libs/partners/anthropic/langchain_anthropic/": "libs/partners/anthropic/tests/
        unit_tests/",
        "libs/partners/chroma/langchain_chroma/": "libs/partners/chroma/tests/unit_tests
        /",
        "libs/partners/exa/langchain_exa/": "libs/partners/exa/tests/unit_tests/",
        "src/transformers/": "tests/",
        "src/transformers/models/": "tests/models/",
        "src/transformers/benchmark/": "tests/benchmark/",
        "inference_sdk/": "tests/inference_sdk/unit_tests/",

        "inference/core/": "tests/inference/unit_tests/core/",
        "inference/enterprise/": "tests/inference/unit_tests/enterprise/",
        "inference/models/": "tests/inference/unit_tests/models/",
        "inference/core/workflows/": "tests/workflows/unit_tests/"
    }
}
```
No explanations are needed, just output in JSON format and using ``` ```.
```
{
    "repo_name": "langchain",
    "testcase_dir_mapping": {
        "libs/core/langchain_core/": "libs/core/tests/unit_tests/",
```

```
596          "libs/langchain/langchain/": "libs/langchain/tests/unit_tests/",
597          "src/transformers/": "tests/",
598          "inference/": "tests/inference/unit_tests/"
599      }
600  }
601  ```
602
```

Our approach supports cases with multiple root directories, such as repositories such as `langchain`, which contain both source code and embedded packages (e.g., `langchain` and `langchain_core`).

After determining the main test and source directories, we traverse all files within these directories to establish fine-grained mappings between individual test files and their corresponding source files. Once valid mappings are identified, we execute the test files in the environment to verify their usability. Additionally, we record the number of test cases in each test file, which is later used to calculate the AC Rate.

## B  Generation Backbone Model Selection

We utilize GPT-4o to generate structured functional descriptions. As shown in Figure 6, different backbone models exhibit different characteristics in their generated descriptions. Although the descriptions generated by Claude 3.5 are accurate, they tend to be more concise and often omit some implementation details, such as variable names and function names. After comparing the styles of descriptions produced by various models, we selected GPT-4o as the backbone for generation, as it provides accurate and detailed descriptions without being verbose.

Furthermore, we evaluate the performance of different models on development-type tasks using descriptions generated by different backbones. As illustrated in Figure 5 the absolute scores of the models fluctuate due to differences in the description styles. However, the relative ranking of the models remains largely consistent and is not affected by the choice of the backbone model.

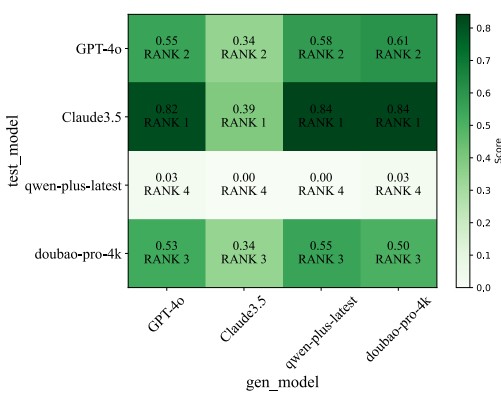

Figure 5: **Performance heatmap between different backbone models and test models**.

Figure 6: Comparison of structured functional descriptions generated by GPT-4o and Claude 3.5.

## C  Prompts for Development Problem Generation

**Prompt for Explanation Generation**

```
634  Please analyze the provided code block based on its context, and output its functionality
635  using concise language in the given format (do not include extra content):
```

```
636  1. **Purpose**
637     Describe the main goal of the code block and its role within the entire program.
638     Specifically, what is its responsibility within the current function?
639  2. **Logic**
640     Elaborate on the core logic and operational process of the code block. For all
641     conditional branches (if statements), explain them one by one.
642     If complex variable updates are involved, use Markdown format for formulas to
643     represent these mathematical calculations.
644     If variables from previous sections of the code block are used, try to describe using
645     their variable names, enclosing them in backticks. Functions should be enclosed in
646     backticks as well, and can be in the form `function_name(arguments)` or `function_name
647     `, without causing ambiguity such as `function_name()` which might lead to
648     misunderstanding.
649  3. **Exceptions**
650     If the code block under analysis throws exceptions, explain its exceptional cases and
651     types. If no exceptions are thrown within the code block, state "None."
652  4. **Variable Assignments**
653     Given the variable list, provide the specific significance and role of the computed
654     variable in the code block in list form.
655     If any variables are incorrectly identified or unused in subsequent sections of code,
656     these can be directly removed.
657     If the variable list is missing any modified variable (such as `self.blockid_list.
658     append(block)`), please add it to the list.
659     Variable list: {variable_list}
660
661  ### Sample Output:
662  1. **Purpose**
663     Parse the target string to extract key information. The target string is in the format
664     `blocks = ["blockid1", "blockid2", ...]`. This code block extracts all valid blockids
665     , generating a new list of strings.
666  2. **Logic**
667     Uses regular expressions (re library) to extract blockid list from the target string,
668     then iterates the list, verifies each blockid's existence in the database, and stores
669     them converted to integer type in a new list.
670  3. **Exceptions**
671     - `ValueError`: If the target string has an incorrect format, making it unable to
672     extract a valid blockid list, this exception is thrown.
673  4. **Variable Assignments**
674     - `self.blockid_list`: Stores extracted and validated blockids
675
676  ### Code Block to be Analyzed:
677  {key_block}
678
679  ### Contextual Information of Code Block:
680  {class_code}
```

## Prompt for Refinement

```
684  The code reviewers found the generated code explanation has the following issues:
685  {response}
686
687  Please modify the current code explanation based on the content of the code block and the
688   reviewers' suggestions, and output it according to the specified format, **do not
689  include extra content**.
690  ### Code Block to be Analyzed:
691  {key_block}
692
693  ### Current Code Explanation:
694  {explanation}
695
696  ### Output Requirements:
697  1. **Purpose**
698     Describe the main goal of the code block and its role within the entire program.
699     Specifically, what is its responsibility within the current function?
700  2. **Logic**
701     Elaborate on the core logic and operational process of the code block. For all
702     conditional branches (if statements), explain them one by one.
703     If complex variable updates are involved, use Markdown format for formulas to
704     represent these mathematical calculations.
705     If variables from previous sections of the code block are used, try to describe using
706     their variable names, enclosing them in backticks.
707  3. **Exceptions**
708     If the analyzed code block throws exceptions (using `raise` statements, excluding `
709     except` statements), explain its exceptional cases and types. If no exceptions are
710     thrown within the code block, state "None."
711  4. **Variable Assignments**
712     Using the provided variable list, describe the specific significance and role of the
713     computed variable in the code block in list form.
```

```
714    If there are any erroneously identified variables (e.g., those not used later in the
715    code), you may directly remove these. If the variable list is missing any modified
716    variable (such as `self.blockid_list.append(block)`), please add it to the list.
717
718    ### Sample Output:
719    1. **Purpose**
720       Parse the target string to extract key information. The target string format is `
721       blocks = ["blockid1", "blockid2", ...]`. This code block extracts all valid blockids
722       and generates a new list of strings.
723    2. **Logic**
724       Uses regular expressions (re library) to extract blockid list from the target string,
725       then iterates the list, verifies each blockid's existence in the database, and stores
726       them converted to integer type in a new list.
727    3. **Exceptions**
728       - `ValueError`: If the target string has an incorrect format making it impossible to
729       extract a valid blockid list, this exception is thrown.
730    4. **Variable Assignments**
731       - `self.blockid_list`: Stores extracted and validated blockids.
```

## D   Multi-Function Problem Generation Rules

In addition to the three basic rules described in Section 3.3, we introduce specific rules for each type of problem to better simulate real-world programming scenarios. For the bugfix type, we only allow the combination of single-function bugfix problems, since in practice programmers typically do not implement new code and debug at the same time. For CORECODEBENCH-*Difficult*, we ensure that each problem contains at least one single-function development problem, and the total number of atomic problems $n$ satisfies $n \geq 3$.

## E   Repository Information

Table 5 presents the basic information of the selected repositories. We selected these repositories from the PyPI library and downloaded their latest release versions. The relative paths between source code and test cases, the implementation styles of test cases (including both unittest and pytest), and the way packages are invoked all vary across these repositories. Nevertheless, our CorePipe is robustly adaptable to these differences, automatically generating corresponding testcases and demonstrating strong generalizability and practical applicability.

Table 5: Repository Information.

| Repo | Created Time | Latest Version | Latest Release Time | Github Link | Total Code Lines | Python Files | Test Files | Test Coverage (%) |
|---|---|---|---|---|---|---|---|---|
| transformers | 2019/9/26 | 4.51.3 | 2025/4/14 | /huggingface/transformers | 971,687 | 1,756 | 712 | 40.55 |
| langchain | 2022/10/25 | 0.3.25 | 2025/5/3 | /langchain-ai/langchain | 68,790 | 1,329 | 265 | 19.94 |
| datachain | 2024/6/27 | 0.16.4 | 2025/5/1 | /iterative/datachain/tree/main | 26,777 | 137 | 57 | 41.61 |
| open-iris | 2023/12/14 | 1.5.0 | 2025/4/22 | /worldcoin/open-iris | 8,072 | 76 | 64 | 84.21 |
| UniRef | 2023/12/26 | 0.6 | 2023/12/26 | /FoundationVision/UniRef | 36,127 | 152 | 50 | 32.89 |
| haystack | 2023/11/25 | 2.13.1 | 2025/4/24 | /deepset-ai/haystack | 33,905 | 211 | 150 | 71.09 |
| d3rlpy | 2020/7/31 | 2.8.1 | 2025/3/2 | /takuseno/d3rlpy | 23,984 | 125 | 45 | 36.00 |
| inference | 2023/8/16 | 0.48.3 | 2025/5/6 | /roboflow/inference | 83,164 | 640 | 118 | 18.44 |
| rdt | 2018/8/23 | 1.16.0 | 2025/4/11 | /sdv-dev/RDT | 7,265 | 31 | 16 | 51.61 |
| cloudnetpy | 2019/9/13 | 1.75.0 | 2025/5/2 | /actris-cloudnet/cloudnetpy | 23,025 | 116 | 49 | 42.24 |
| skfolio | 2023/12/15 | 0.9.0 | 2025/4/5 | /skfolio/skfolio | 29,865 | 113 | 71 | 62.83 |
| finam | 2023/2/3 | 1.0.1 | 2025/4/23 | /finam-ufz/finam | 12,592 | 46 | 30 | 65.22 |

## F   Data Source Illustration of CORECODEBENCH

**Problem Statement Examples.** In Figure 7, we present the four atomic types of single-function problems. All problem types are constructed by rewriting or masking candidate key blocks, ensuring that the code under test remains both core and complete. The multi-function problems are generated by combining single-function problems, and their descriptions and formats are consistent with those of the single-function problems.

## G   Prompts of Evaluation

Below, we present the evaluation prompts used for each of the six problem types.

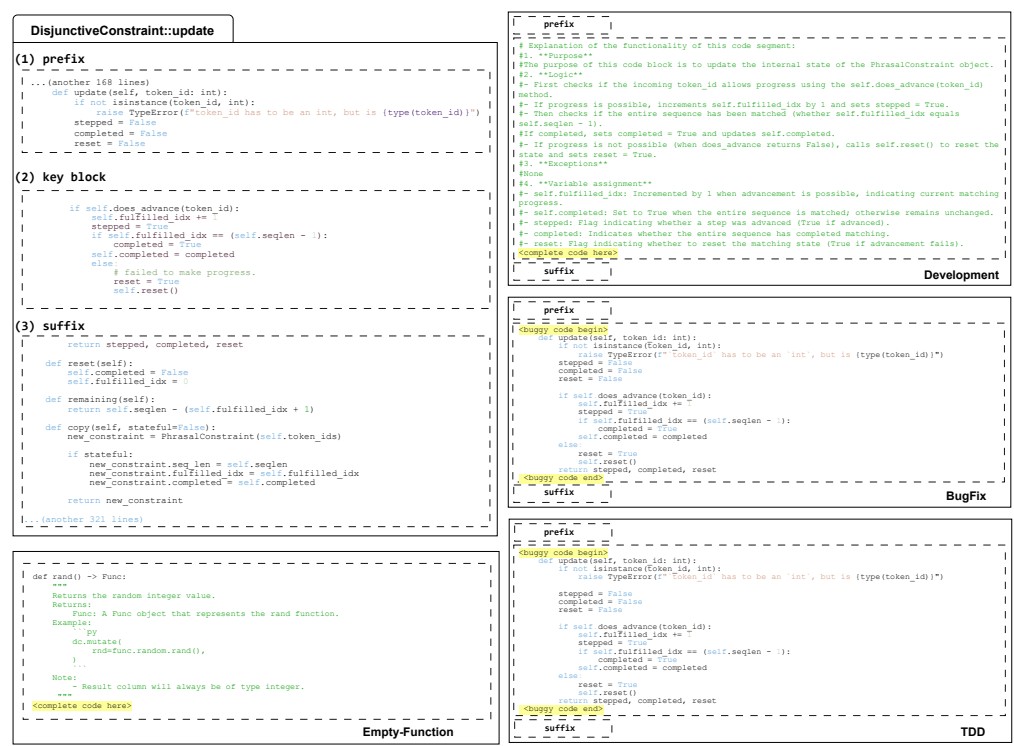

Figure 7: Illustration of atomic single-function problems.

## G.1 Single-Function Evaluation

### G.1.1 Development

```
Below is a code snippet containing a placeholder `<complete code here>`. Please analyze
the provided context and description of the missing code to generate the appropriate code
 block at `<complete code here>`.
Please output the completed code block using markdown format (```python```).
**Important**: Ensure the code block you complete maintains the same indentation as the
context code, meaning you need to preserve the original code's indentation.The output
must exactly match the line count and structure of the input, including preserving empty
lines and comment positions.
Code snippet:
```python
{prompt}
```
Please output the completed code block using markdown format (```python```). Make sure to
 preserve the original indentation before and after the <complete code here> placeholder.
 And remember don't add the signature of the function into it.
```

### G.1.2 BugFix

```
In the following code snippet, there is a buggy code section between `<buggy code begin>`
 and `<buggy code end>`. I've provided the corresponding unit test file and pytest error
messages. Please analyze the given context and rewrite the erroneous code segment.
Please format the rewritten function block in markdown (```python```), including only the
 rewritten content between `<buggy code begin>` and `<buggy code end>`, without including
 the `<buggy code begin>` and `<buggy code end>` tags.
**Note**: Please ensure that your completed code block maintains the indentation of the
original code context.
Code snippet:
```python
{new_code}
```
Unit test code:
```python
{test_code}
```

```
791 ```
792 Test error log:
793 ```
794 {log}
795 ```
796
```

### G.1.3 TDD

```
799 Below is a code file {file_name} containing a placeholder `<complete code here>`.Please
800 analyze the provided file context and unit test information, and generate appropriate
801 code at the `<complete code here>` location. Please output your completed code block in
802 markdown format (```python```). The code block should only include the code at the `<
803 completed code here>` location, without the surrounding context.
804 **Note**: Please ensure that your completed code block maintains the indentation of the
805 surrounding code, meaning you need to preserve the original code's indentation.
806
807 Code file {file_name} to be completed:
808 ```python
809 {new_code}
810 ```
811 Corresponding unit test:
812 ```python
813 {test_file}
814 ```
```

## G.2 Multi-Function Evaluation

### G.2.1 Development

```
819 You are a code completion agent, I would provide you with a snippet of code, and you
820 would need to return the completed code segment.
821 the code after <ralated code> is used while calling the code to be completed.
822 You need to complete code blocks after <complete following code> by predicting the codes
823 after <complete code here>, <id> label wraps the position of the code.
824 Your output should include the <id></id> label, followed by the completed code snippet
825 enclosed within triple backticks ```, ensuring clarity and proper formatting.
826
827 <related code>
828 <id>{id}<\id>
829 {related code}
830
831 <complete following code>
832 <id>{id}<\id>
833 {function code}
834
```

### G.2.2 BugFix

```
837 In the following code snippet, the code between <buggy code begin> and <buggy code end>
838 contains bugs, <id> label wraps the position of the code. Please analyze the provided
839 context and rewrite the faulty code segment.
840 The code after <ralated code> is used while calling the code to be rewrited.
841 Your output should include the <id></id> label, followed by the new code snippet enclosed
842  within triple backticks ```, ensuring clarity and proper formatting.
843
844 <related code>
845 <id>{id}<\id>
846 {related code}
847
848 <complete following code>
849 <id>{id}<\id>
850 {function code}
851
```

### G.2.3 TDD

```
854 You are a code completion agent, I would provide you with a snippet of code, and you
855 would need to return the completed code segment.
856 The code after <ralated code> is used while calling the code to be completed.
857 You need to complete code blocks after <complete following code> by predicting the codes
858 after <complete code here>, <id> label wraps the position of the code.
859 Please analyze the provided file context and the unit test information of the file, and
860 generate an appropriate code block at the position marked <complete code here>.
```

```
861  Your output should include the <id></id> label, followed by the completed code snippet
862  enclosed within triple backticks ```, ensuring clarity and proper formatting.
863  Note: Please ensure that the code block you provide as a completion matches the
864  indentation of the surrounding context, i.e., you need to preserve the original code's
865  indentation.
866
867  <related code>
868  <id>{id}<\id>
869  {related code}
870
871  <complete following code>
872  <id>{id}<\id>
873  {function code}
874
875  The unit test information:
876  {test_codes}
877
```

## H  Human Annotator

We employ three annotators in total, all of whom hold bachelor's degrees or higher in computer-related fields (including information security and software engineering) and have at least two years of Python development experience.

The annotators work full-time, with a daily workload not exceeding 8 hours. Their annotation productivity is approximately 1 task per hour, and they are compensated following local labor regulations.

Furthermore, we conduct quality checks on their annotations. All quality inspectors are members of the author team with educational backgrounds in computer science and artificial intelligence.

## I  Implementation Details

For the selection of key blocks, we require that each key block contains more than 10 lines of code to ensure the difficulty of the problems. In multi-function problem generation, we set $\nu = 6, d = 3$. For CORECODEBENCH-*Difficult*, we use $\nu = \infty, d = 3$. The depth $d$ is set to 3 to better simulate real-world development scenarios.

During the evaluation, we extract the model output using regular expressions. To reduce bias caused by format mismatches in model output, we further apply post-processing steps such as indentation alignment and function header alignment, ensuring that the outputs are as evaluable as possible.

## J  Detailed Results of Single-Function Problems

In Table 6, Table 7, and Table 8, we present the detailed evaluation results of single-function problems across different repositories and types of problems. It should be noted that `langchain` and `langchain_core` are listed separately in the tables, as their source directories, test file locations, and runtime Python paths are different.

## K  Detailed Results of Multi-Function Problems

In Table 9, Table 10, and Table 11, we present the detailed evaluation results of multi-function problems across different repositories and types of problems. Similar to Appendix J, we separate `langchain` and `langchain_core` as different repositories.

Table 6: Detailed results of single-function development problems.

| model/repo | transformers | | langchain | | langchain_core | | datachain | | open-ins | | UniRef | | haystack | | d3rlpy | | inference | | rdt | | cloudnetpy | | sk-folio | | finam | |
|---|---|---|---|---|---|---|---|---|---|---|---|---|---|---|---|---|---|---|---|---|---|---|---|---|---|---|---|
| | AC Rate | AC@1 | AC Rate | AC@1 | AC Rate | AC@1 | AC Rate | AC@1 | AC Rate | AC@1 | AC Rate | AC@1 | AC Rate | AC@1 | AC Rate | AC@1 | AC Rate | AC@1 | AC Rate | AC@1 | AC Rate | AC@1 | AC Rate | AC@1 | AC Rate | AC@1 |
| GPT-4o | 81.88 | 48.83 | 75.45 | 71.88 | 93.22 | 61.53 | 82.38 | 63.79 | 80.09 | 38.46 | 92.46 | 62.50 | 81.35 | 39.80 | 62.58 | 47.06 | 89.17 | 48.15 | 96.95 | 46.15 | 65.34 | 45.45 | 73.83 | 48.94 | 75.65 | 57.14 |
| GPT4.1 | 82.84 | 53.49 | 80.33 | 84.38 | 93.31 | 61.54 | 86.90 | 70.69 | 84.28 | 53.85 | 94.01 | 81.25 | 85.88 | 45.92 | 67.18 | 58.82 | 89.62 | 58.82 | 93.57 | 29.63 | 68.57 | 45.45 | 71.72 | 53.19 | 77.12 | 57.14 |
| o1-mini | 84.94 | 43.02 | 69.88 | 56.25 | 92.84 | 61.64 | 88.32 | 63.79 | 66.17 | 23.08 | 85.53 | 50.00 | 71.78 | 32.65 | 52.90 | 38.24 | 48.83 | 23.33 | 92.59 | 25.93 | 65.34 | 45.45 | 85.29 | 48.94 | 67.12 | 28.57 |
| o4-mini | 88.48 | 60.47 | 81.92 | 75.00 | 96.10 | 69.23 | 90.43 | 72.41 | 92.63 | 46.15 | 97.14 | 75.00 | 80.35 | 37.76 | 83.23 | 63.33 | 89.40 | 63.33 | 93.76 | 37.04 | 67.15 | 41.86 | 80.06 | 48.94 | 85.88 | 57.14 |
| Claude-3.5-Sonnet | 82.89 | 55.81 | 91.67 | 90.63 | 93.27 | 53.85 | 85.73 | 65.51 | 87.07 | 46.15 | 96.63 | 75.00 | 80.66 | 34.69 | 89.83 | 56.67 | 90.37 | 56.67 | 97.68 | 44.44 | 67.82 | 40.91 | 77.45 | 46.81 | 85.04 | 50.00 |
| Claude-3.7-Sonnet | 88.64 | 62.79 | 93.45 | 90.63 | 95.09 | 61.54 | 84.54 | 68.97 | 85.26 | 38.46 | 96.54 | 87.50 | 84.63 | 40.82 | 89.17 | 58.82 | 89.12 | 56.67 | 96.47 | 44.44 | 70.39 | 50.00 | 76.73 | 51.06 | 75.82 | 50.00 |
| Gemini-2.5-Pro-Preview | 82.27 | 55.81 | 76.00 | 65.62 | 68.32 | 46.15 | 83.01 | 62.07 | 60.56 | 30.77 | 83.59 | 68.75 | 71.05 | 33.67 | 85.76 | 67.65 | 87.51 | 53.33 | 84.19 | 29.63 | 57.26 | 37.21 | 71.93 | 38.30 | 69.20 | 50.00 |
| Grok-3 | 84.37 | 60.47 | 64.62 | 59.38 | 77.12 | 60.00 | 80.49 | 58.62 | 84.65 | 27.27 | 76.30 | 56.25 | 78.06 | 31.63 | 92.58 | 76.47 | 83.39 | 76.47 | 87.08 | 44.44 | 65.08 | 44.19 | 87.97 | 65.96 | 79.05 | 50.00 |
| Douhao-pro-4k | 78.44 | 36.05 | 82.74 | 71.88 | 46.15 | 72.97 | 79.74 | 56.90 | 75.80 | 26.70 | 80.26 | 83.43 | 72.79 | 25.51 | 53.44 | 35.29 | 73.15 | 30.00 | 96.46 | 33.33 | 64.56 | 36.36 | 34.04 | 0 | 42.86 | 68.33 |
| Douhao-1.5pro | 85.83 | 55.81 | 82.72 | 75.00 | 92.78 | 53.85 | 88.77 | 65.52 | 96.06 | 69.23 | 88.33 | 75.00 | 77.91 | 38.78 | 73.61 | 38.78 | 92.52 | 63.33 | 97.87 | 51.85 | 65.80 | 51.16 | 79.51 | 48.94 | 73.06 | 42.86 |
| qwen-plus-latest | 79.44 | 45.35 | 81.32 | 71.88 | 92.54 | 53.85 | 83.11 | 63.79 | 80.10 | 46.15 | 91.48 | 68.75 | 80.36 | 35.71 | 79.76 | 37.76 | 88.07 | 53.33 | 93.07 | 37.04 | 60.14 | 31.82 | 70.24 | 42.55 | 73.39 | 42.86 |
| Qwen2.5-max | 85.03 | 61.62 | 82.07 | 71.88 | 94.85 | 61.54 | 82.90 | 62.07 | 80.88 | 46.15 | 99.35 | 90.91 | 80.14 | 37.76 | 67.25 | 35.71 | 83.38 | 50.00 | 91.61 | 40.74 | 65.14 | 40.91 | 78.03 | 46.81 | 74.94 | 42.86 |
| DeepSeek-Coder-V2-Lite-Instruct-16B | 70.79 | 9.30 | 35.53 | 21.88 | 67.55 | 15.38 | 75.48 | 36.21 | 65.64 | 15.38 | 67.22 | 18.75 | 66.16 | 14.29 | 63.25 | 14.29 | 68.23 | 11.11 | 91.69 | 11.11 | 47.47 | 6.98 | 68.11 | 17.02 | 55.96 | 7.14 |
| DeepSeek-R1 | 87.11 | 61.63 | 91.67 | 90.63 | 95.65 | 69.23 | 75.48 | 36.21 | 73.90 | 46.15 | 92.52 | 68.75 | 81.74 | 41.84 | 88.40 | 73.53 | 89.85 | 73.53 | 97.32 | 40.74 | 62.74 | 41.86 | 79.54 | 53.19 | 74.73 | 50.00 |
| Llama3.1-70B | 66.21 | 23.26 | 64.96 | 56.25 | 94.01 | 69.23 | 78.59 | 55.17 | 68.05 | 23.08 | 68.71 | 43.75 | 50.20 | 12.24 | 54.33 | 32.35 | 74.34 | 46.67 | 89.40 | 7.41 | 67.15 | 31.82 | 75.76 | 40.43 | 75.93 | 48.82 |
| Qwen3-8B | 64.27 | 3.49 | 43.28 | 18.75 | 57.46 | 0 | 41.46 | 6.90 | 55.21 | 7.69 | 44.84 | 0 | 50.75 | 9.18 | 47.39 | 20.59 | 60.73 | 20.00 | 83.00 | 7.41 | 38.14 | 4.65 | 60.93 | 8.51 | 49.58 | 0 |

Table 7: Detailed results of single-function bugfix problems.

| model/repo | transformers | | langchain | | langchaincore | | datachain | | UniRef | | haystack | | d3rlpy | | inference | | rdt | | cloudnetpy | | sk-folio | | finam | |
|---|---|---|---|---|---|---|---|---|---|---|---|---|---|---|---|---|---|---|---|---|---|---|---|---|
| | AC Rate | AC@1 | AC Rate | AC@1 | AC Rate | AC@1 | AC Rate | AC@1 | AC Rate | AC@1 | AC Rate | AC@1 | AC Rate | AC@1 | AC Rate | AC@1 | AC Rate | AC@1 | AC Rate | AC@1 | AC Rate | AC@1 | AC Rate | AC@1 |
| GPT4o | 62.87 | 29.51 | 70.96 | 57.57 | 45.95 | 33.33 | 77.5 | 44.64 | 43.51 | 22.22 | 59.63 | 20.77 | 75.93 | 57.5 | 82.62 | 60 | 68.96 | 37.5 | 39.95 | 25 | 39.28 | 16.67 | 40.91 | 8.33 |
| GPT4.1 | 71.36 | 37.71 | 80.33 | 66.67 | 58.91 | 50 | 84.13 | 53.57 | 62.96 | 44.44 | 69.92 | 40.26 | 89.66 | 77.5 | 92.63 | 75 | 79 | 58.33 | 56.95 | 40.62 | 64.58 | 50 | 52.03 | 16.67 |
| O1-mini | 67.54 | 22.95 | 73.92 | 54.54 | 38.96 | 25 | 77.58 | 48.21 | 35.18 | 22.22 | 57.04 | 19.48 | 78.81 | 65 | 59.92 | 35 | 74.59 | 33.33 | 38.24 | 28.12 | 44.08 | 30 | 41.55 | 8.3 |
| O4-mini | 73.1 | 42.62 | 73.91 | 54.54 | 45.33 | 40 | 85.57 | 66.07 | 75.92 | 55.55 | 51.94 | 51.94 | 91.5 | 85 | 71.77 | 35 | 85.23 | 66.66 | 46.6 | 43.72 | 57.73 | 33.33 | 52.25 | 33.33 |
| Claude3.5 | 64.97 | 27.87 | 75.78 | 60.6 | 38.05 | 25 | 78.43 | 50 | 68.51 | 55.55 | 60.93 | 23.37 | 87.73 | 77.5 | 55 | 30 | 70.71 | 37.5 | 39.96 | 34.37 | 57.27 | 38.89 | 68.24 | 25 |
| Claude3.7 | 64.87 | 37.7 | 82.06 | 69.69 | 39.81 | 33.33 | 84.39 | 57.14 | 51.85 | 33.33 | 64.98 | 29.87 | 81.5 | 75 | 71.77 | 35 | 76.72 | 58.33 | 43.52 | 34.37 | 51.28 | 33.33 | 63.44 | 25 |
| gemini2.5-pro | 30.63 | 19.67 | 39.43 | 36.36 | 26 | 20 | 44.91 | 41.07 | 27.16 | 11.11 | 26.28 | 14.28 | 38.42 | 32.5 | 54.01 | 40 | 21.91 | 12.5 | 27.85 | 21.87 | | | 2.08 | 0 |
| Grok3 | 38.9 | 19.67 | 82.25 | 63.63 | 17.04 | 8.33 | 78.54 | 39.28 | 46.29 | 33.33 | 22.08 | 22.08 | 74.15 | 60 | 63.22 | 40 | 69.22 | 41.67 | 49.8 | 37.5 | 35.64 | 16.67 | 41.2 | 25 |
| douhao4kPro | 66.23 | 31.15 | 77.49 | 57.57 | 35.02 | 33.33 | 82.22 | 46.43 | 46.29 | 33.33 | 27.27 | 27.27 | 85.88 | 75 | 88.08 | 60 | 71.74 | 41.67 | 44.04 | 31.25 | 44.74 | 27.78 | 53.03 | 8.33 |
| douhao-1.5-pro | 67.29 | 31.15 | 73.42 | 54.54 | 58.3 | 50 | 82.22 | 48.21 | 54.62 | 33.33 | 63.53 | 19.48 | 86.37 | 77.5 | 88.07 | 60 | 73.37 | 41.67 | 40.81 | 31.25 | 47.39 | 33.33 | 44.32 | 16.67 |
| deepseek-16B-Coder-V2-Lite-Instruct | 45.12 | 13.11 | 46.24 | 24.24 | 25 | 25 | 48.84 | 21.43 | 0 | 0 | 2.6 | 2.6 | 36.23 | 25 | 13.33 | 10 | 30.24 | 9.37 | 15.59 | 9.37 | 30.45 | 16.67 | 8.7 | 0 |
| deepseekR1 | 68.89 | 32.79 | 80.63 | 69.7 | 58.3 | 58.3 | 82.38 | 57.14 | 56.48 | 22.22 | 24.67 | 24.67 | 86.15 | 72.5 | 83.32 | 65 | 70.6 | 37.5 | 45.88 | 34.37 | 59.16 | 50 | 48.78 | 25 |
| Llama3.1-70B | 59.02 | 21.31 | 74.37 | 57.57 | 34.09 | 33.33 | 70.53 | 41.07 | 41.04 | 11.11 | 51.64 | 15.58 | 80.44 | 70 | 39.09 | 25 | 56.14 | 20.83 | 43.38 | 31.25 | 33.69 | 16.66 | 39.77 | 0 |
| qwen-plus-latest | 48.8 | 22.95 | 54.94 | 42.42 | 28.7 | 25 | 71.86 | 33.93 | 11.11 | 11.11 | 20.78 | 20.78 | 54.14 | 42.5 | 48.19 | 40 | 40.94 | 12.5 | 27.42 | 18.75 | 32.11 | 16.67 | 24.17 | 0 |
| Qwen2.5-max | 58.28 | 26.23 | 75.67 | 51.51 | 28.7 | 25 | 73.18 | 41.07 | 24.07 | 11.11 | 57.44 | 18.18 | 68.87 | 57.5 | 63.54 | 50 | 55.56 | 25 | 39.78 | 21.87 | 45.73 | 27.78 | 42.18 | 0 |
| qwen3-8B | 38.84 | 6.55 | 40.77 | 15.15 | 0 | 0 | 56.7 | 23.21 | 19.57 | 0 | 1.3 | 1.3 | 29.86 | 7.5 | 0 | 0 | 34.84 | 0 | 17.84 | 9.37 | 20.27 | 11.11 | 10.99 | 0 |
| Qwen3-32B/ | 53.37 | 19.67 | 61.56 | 45.45 | 33.33 | 25 | 60.55 | 33.93 | 22.22 | 22.22 | 32.15 | 6.49 | 58.31 | 45 | 24.03 | 15 | 40.83 | 8.3 | 26.76 | 18.75 | | | 19.82 | 8.33 |

Table 8: Detailed results of single-function TDD problems.

| model/repo | transformers | | langchaincore | | datachain | | open-iris | | haystack | | cloudnetpy | | skfolio | | finam | |
|---|---|---|---|---|---|---|---|---|---|---|---|---|---|---|---|---|
| | AC Rate | AC @1 | AC Rate | AC @1 | AC Rate | AC @1 | AC Rate | AC @1 | AC Rate | AC @1 | AC Rate | AC @1 | AC Rate | AC @1 | AC Rate | AC @1 |
| GPT4o | 85.29 | 50.7 | 79.71 | 44.19 | 75.69 | 46.43 | 82.18 | 36.36 | 84.05 | 38.89 | 90.47 | 52.5 | 79.49 | 46.42 | 95.86 | 55.55 |
| GPT4.1 | 90.26 | 66.2 | 89.94 | 72.09 | 85.24 | 50 | 84.67 | 45.45 | 81.32 | 66.66 | 93.59 | 75 | 91.17 | 67.85 | 92.28 | 44.44 |
| O1-mini | 84.12 | 46.48 | 84.38 | 58.13 | 77.88 | 53.57 | 85.89 | 54.54 | 80.09 | 50 | 90.25 | 62.5 | 84.82 | 57.14 | 95.86 | 55.55 |
| O4-mini | 87.03 | 61.97 | 79.07 | 69.77 | 80.26 | 67.86 | 82.61 | 54.54 | 88.88 | 83.33 | 95.07 | 75 | 90.15 | 71.43 | 93.99 | 77.78 |
| Claude3.5 | 90.27 | 64.79 | 73.61 | 46.51 | 84.23 | 60.71 | 94.03 | 63.63 | 69.76 | 61.11 | 95.78 | 67.5 | 86.26 | 53.57 | 93.13 | 66.66 |
| Claude3.7 | 88.43 | 64.79 | 79.93 | 60.46 | 83.33 | 60.71 | 77.14 | 36.36 | 82.01 | 66.66 | 95.7 | 80 | 83.55 | 55.35 | 93.98 | 66.66 |
| gemini2.5-pro | 82.74 | 57.75 | 68.86 | 55.81 | 67.41 | 53.57 | 72.3 | 36.4 | 82.8 | 55.6 | 81.9 | 57.5 | 63.5 | 41.1 | 76.2 | 55.6 |
| Grok3 | 73.67 | 46.48 | 81.62 | 60.46 | 83.9 | 53.57 | 84.37 | 45.45 | 85.07 | 61.11 | 91.48 | 60 | 87.42 | 58.93 | 93.13 | 66.66 |
| doubao4kPro | 78.12 | 33.8 | 70.5 | 32.56 | 62.81 | 25 | 72.5 | 18.18 | 71.31 | 16.66 | 79.79 | 32.5 | 81.36 | 35.71 | 92.44 | 55.55 |
| doubao 1.5 pro | 85.27 | 52.11 | 84.24 | 51.16 | 74.06 | 39.29 | 82.56 | 36.36 | 71.31 | 16.66 | 88.08 | 57.5 | 84.67 | 55.35 | 95.86 | 55.55 |
| deepseek-16B-Coder-V2-Lite-Instruct | 60.13 | 21.13 | 36.96 | 11.63 | 57.91 | 17.86 | 55.27 | 18.18 | 85.83 | 55.55 | 64.37 | 17.5 | 70.5 | 25 | 96.72 | 66.67 |
| deepseekR1 | 87.45 | 64.79 | 80.63 | 55.81 | 79.34 | 46.43 | 85.44 | 63.63 | 52.95 | 16.66 | 95.95 | 75 | 88.3 | 64.28 | 95.01 | 44.44 |
| Llama3.1-70B | 79.93 | 35.21 | 64.43 | 30.23 | 73.56 | 39.29 | 83.9 | 45.45 | 71.55 | 22.22 | 88.17 | 42.5 | 78.78 | 39.29 | 95.86 | 55.55 |
| qwen-plus-latest | 80.9 | 47.89 | 71.55 | 39.53 | 71.23 | 39.28 | 75.53 | 18.18 | 82.63 | 22.22 | 87.76 | 52.5 | 81.6 | 42.86 | 92.28 | 55.55 |
| Qwen2.5-max | 85.3 | 50.7 | 79.49 | 37.21 | 72.34 | 35.71 | 87.19 | 54.54 | 76.05 | 50 | 88.37 | 47.5 | 81.65 | 50 | 85.84 | 22.22 |
| qwen3-8B | 68.41 | 14.08 | 38.83 | 11.63 | 35.96 | 35.96 | 64.8 | 9.09 | 56.8 | 22.22 | 62.65 | 20 | 66.45 | 16.07 | | |

Table 9: Detailed results of multi-function development problems.

| model/repo | transformers | | langchain | | langchaincore | | datachain | | open-iris | | UniRef | | haystack | | d34py | | inference | | ndt | | cloudnetpy | | skfolio | | finam | |
|---|---|---|---|---|---|---|---|---|---|---|---|---|---|---|---|---|---|---|---|---|---|---|---|---|---|---|---|---|
| | AC Rate | AC @1 | AC Rate | AC @1 | AC Rate | AC @1 | AC Rate | AC @1 | AC Rate | AC @1 | AC Rate | AC @1 | AC Rate | AC @1 | AC Rate | AC @1 | AC Rate | AC @1 | AC Rate | AC @1 | AC Rate | AC @1 | AC Rate | AC @1 | AC Rate | AC @1 |
| GPT-4o | 24.57 | 0 | 7.29 | 0 | 30.46 | 16.67 | 34.69 | 12.50 | 6.67 | 0 | 20.67 | 0 | 16.99 | 0 | 3.85 | 0 | 36.09 | 0 | 28.79 | 16.67 | 0 | 0 | 14.40 | 3.85 | 0 | 0 |
| GPT-4.1 | 14.33 | 7.14 | 10.71 | 0 | 16.67 | 16.67 | 32.93 | 0 | 25.56 | 10.00 | 26.00 | 0 | 11.40 | 4.65 | 3.85 | 0 | 3.00 | 0 | 12.12 | 0 | 2.78 | 0 | 4.56 | 0 | 0 | 0 |
| o1-mini | 9.75 | 0 | 13.54 | 0 | 17.36 | 0 | 28.75 | 0 | 3.33 | 0 | 6.67 | 0 | 23.12 | 4.65 | 5.77 | 0 | 24.65 | 0 | 45.96 | 16.67 | 9.26 | 0 | 18.98 | 11.54 | 14.10 | 0 |
| o4-mini | 21.60 | 0 | 11.76 | 0 | 18.15 | 16.67 | 37.07 | 0 | 35.87 | 20.00 | 22.67 | 0 | 22.60 | 6.98 | 5.77 | 0 | 11.06 | 0 | 49.14 | 16.67 | 3.03 | 0 | 16.83 | 15.38 | 14.10 | 0 |
| Claude-3.5-Sonnet | 29.94 | 14.29 | 43.90 | 25.00 | 41.80 | 16.67 | 35.66 | 0 | 11.11 | 0 | 30.33 | 0 | 1.99 | 0 | 57.12 | 25.00 | 46.29 | 0 | 37.32 | 0 | 5.56 | 0 | 23.79 | 11.54 | 2.78 | 0 |
| Claude-3.7-Sonnet | 52.49 | 28.57 | 47.17 | 12.50 | 51.06 | 33.33 | 35.54 | 0 | 16.67 | 0 | 43.67 | 0 | 29.47 | 4.65 | 11.54 | 0 | 31.08 | 0 | 52.47 | 16.67 | 12.04 | 8.33 | 20.75 | 3.85 | 14.10 | 0 |
| Gemini-2.5-Pro-Preview | 26.88 | 0 | 18.75 | 12.50 | 29.33 | 16.67 | 32.54 | 0 | 6.67 | 0 | 25.00 | 0 | 15.23 | 4.65 | 16.35 | 0 | 38.82 | 0 | 55.51 | 16.67 | 5.56 | 0 | 13.21 | 2.56 | 15.38 | 0 |
| Grok-3 | 20.00 | 7.14 | 37.65 | 25.00 | 47.52 | 33.33 | 33.32 | 12.50 | 16.67 | 0 | 35.00 | 0 | 15.92 | 4.65 | 0 | 0 | 38.42 | 0 | 16.67 | 0 | 32.66 | 25.00 | 22.04 | 0 | 0 | 0 |
| Douban-pro-4k | 5.09 | 0 | 0 | 0 | 0 | 0 | 27.81 | 0 | 6.67 | 0 | 0 | 0 | 6.87 | 0 | 0 | 0 | 9.38 | 0 | 0 | 0 | 0 | 0 | 0 | 0 | 0 | 0 |
| Douban-1.5-pro | 5.09 | 0 | 0 | 0 | 0 | 0 | 27.81 | 0 | 0 | 0 | 0 | 0 | 6.77 | 0 | 5.77 | 0 | 0 | 0 | 30.30 | 16.67 | 0 | 0 | 20.90 | 11.54 | 14.10 | 0 |
| qwen-plus-latest | 30.97 | 14.29 | 12.50 | 0 | 28.70 | 16.67 | 25.60 | 0 | 24.76 | 10.00 | 22.33 | 0 | 17.94 | 6.98 | 29.19 | 0 | 20.13 | 0 | 43.23 | 16.67 | 21.55 | 16.67 | 18.72 | 15.38 | 2.56 | 0 |
| Qwen2.5-max | 32.65 | 21.43 | 18.75 | 12.50 | 34.13 | 25.00 | 39.08 | 12.50 | 23.33 | 10.00 | 14.83 | 0 | 24.81 | 6.98 | 0 | 0 | 18.33 | 0 | 0 | 0 | 4.63 | 0 | 0 | 0 | 0 | 0 |
| DeepSeek-Coder-V2-Lite-Instruct-16B | 0 | 0 | 0 | 0 | 0 | 0 | 0 | 0 | 0 | 0 | 0 | 0 | 0 | 0 | 31.11 | 0 | 0 | 0 | 19.70 | 0 | 0 | 0 | 23.96 | 15.38 | 2.56 | 0 |
| DeepSeek-R1 | 32.62 | 7.14 | 31.25 | 25.00 | 20.13 | 8.33 | 41.90 | 0 | 23.33 | 10.00 | 26.33 | 0 | 27.89 | 4.65 | 7.69 | 0 | 17.25 | 0 | 49.77 | 16.67 | 2.78 | 0 | 20.38 | 7.69 | 0 | 0 |
| Llama3.1-70B | 43.84 | 21.43 | 18.01 | 0 | 19.35 | 8.33 | 27.62 | 0 | 9.44 | 0 | 8.33 | 0 | 17.61 | 2.33 | 0 | 0 | 9.83 | 0 | 0 | 0 | 12.04 | 8.33 | 20.38 | 0 | 0 | 0 |
| Qwen3-8B | 0 | 0 | 0 | 0 | 0 | 0 | 0 | 0 | 0 | 0 | 0 | 0 | 0 | 0 | | | 0 | 0 | 0 | 0 | 0 | 0 | | | 0 | 0 |

Table 10: Detailed results of multi-function bugfix problems.

| model/repo | transformers | | langchain | | datachain | | haystack | | d3rlpy | |
|---|---|---|---|---|---|---|---|---|---|---|
| | AC Rate | AC @1 | AC Rate | AC @1 | AC Rate | AC @1 | AC Rate | AC @1 | AC Rate | AC @1 |
| GPT-4o | 69.23 | 0 | 0 | 0 | 28.28 | 0 | 8.33 | 0 | 0 | 0 |
| GPT-4.1 | 69.23 | 0 | 100 | 100 | 27.27 | 0 | 8.33 | 0 | 15.71 | 0 |
| o1-mini | 69.23 | 0 | 100 | 100 | 27.27 | 0 | 0 | 0 | 10.71 | 0 |
| o4-mini | 69.23 | 0 | 100 | 100 | 28.28 | 0 | 0 | 0 | 15.71 | 0 |
| Claude-3.5-Sonnet | 69.23 | 0 | 100 | 100 | 27.27 | 0 | 0 | 0 | 10.71 | 0 |
| Claude-3.7-Sonnet | 69.23 | 0 | 100 | 100 | 28.28 | 0 | 0 | 0 | 10.71 | 0 |
| Gemini-2.5-Pro-Preview | 0 | 0 | 0 | 0 | 0 | 0 | 0 | 0 | 10.71 | 0 |
| Grok-3 | 69.23 | 0 | 0 | 0 | 0 | 0 | 8.33 | 0 | 0 | 0 |
| Doubao-pro-4k | 69.23 | 0 | 100 | 100 | 27.27 | 0 | 0 | 0 | 2.50 | 0 |
| Doubao-1.5-pro | 69.23 | 0 | 0 | 0 | 2.02 | 0 | 0 | 0 | 10.71 | 0 |
| qwen-plus-latest | 69.23 | 0 | 100 | 100 | 0 | 0 | 0 | 0 | 0 | 0 |
| Qwen2.5-max | 100 | 100 | 0 | 0 | 27.27 | 0 | 8.33 | 0 | 10.71 | 0 |
| DeepSeek-Coder-V2-Lite-Instruct-16B | 0 | 0 | 0 | 0 | 0 | 0 | 0 | 0 | 0 | 0 |
| DeepSeek-R1 | 69.23 | 0 | 100 | 100 | 0 | 0 | 0 | 0 | 10.71 | 0 |
| Llama3.1-70B | 69.23 | 0 | 0 | 0 | 0 | 0 | 8.33 | 0 | 10.71 | 0 |
| Qwen3-8B | 69.23 | 0 | 0 | 0 | 0 | 0 | 0 | 0 | 0 | 0 |
| qwen3-32b-no-thinking | 0 | 0 | 0 | 0 | 0 | 0 | 0 | 0 | 0 | 0 |

Table 11: Detailed results of multi-function TDD problems.

| model/repo | transformers | | langchain.core | | datachain | | open-iris | | haystack | | inference | | cloudnetpy | | skfolio | | finam | |
|---|---|---|---|---|---|---|---|---|---|---|---|---|---|---|---|---|---|---|
| | AC Rate | AC @1 | AC Rate | AC @1 | AC Rate | AC @1 | AC Rate | AC @1 | AC Rate | AC @1 | AC Rate | AC @1 | AC Rate | AC @1 | AC Rate | AC @1 | AC Rate | AC @1 |
| GPT-4o | 22.77 | 6.25 | 18.10 | 14.29 | 4.10 | 0 | 21.30 | 0 | 14.50 | 6.25 | 43.79 | 16.67 | 15.19 | 6.67 | 14.70 | 11.11 | 10.94 | 0 |
| GPT-4.1 | 11.02 | 0 | 33.39 | 21.43 | 6.18 | 0 | 26.85 | 11.11 | 43.76 | 20.83 | 35.39 | 0 | 13.33 | 13.33 | 10.30 | 7.41 | 20.91 | 0 |
| o1-mini | 32.07 | 12.50 | 12.92 | 7.14 | 5.19 | 0 | 7.41 | 0 | 29.42 | 8.33 | 23.65 | 0 | 11.58 | 0 | 19.27 | 7.41 | 21.51 | 0 |
| o4-mini | 45.72 | 31.25 | 41.95 | 28.57 | 16.31 | 11.11 | 20.11 | 11.11 | 56.75 | 35.42 | 53.40 | 16.67 | 28.49 | 13.33 | 22.22 | 22.22 | 23.44 | 12.50 |
| Claude-3.5-Sonnet | 48.71 | 18.75 | 44.19 | 21.43 | 27.48 | 0 | 28.24 | 11.11 | 44.84 | 16.67 | 11.72 | 0 | 14.54 | 6.67 | 19.01 | 11.11 | 9.38 | 0 |
| Claude-3.7-Sonnet | 56.09 | 25.00 | 46.42 | 35.71 | 9.64 | 0 | 22.42 | 11.11 | 40.44 | 22.92 | 31.67 | 16.67 | 28.15 | 20.00 | 28.67 | 22.22 | 20.91 | 0 |
| Gemini-2.5-Pro-Preview | 24.28 | 12.50 | 16.47 | 7.14 | 23.41 | 0 | 18.52 | 11.11 | 34.96 | 20.83 | 37.47 | 0 | 10.00 | 6.67 | 23.25 | 18.52 | 9.38 | 0 |
| Grok-3 | 26.30 | 12.50 | 21.27 | 7.14 | 39.95 | 11.11 | 7.41 | 0 | 21.06 | 12.50 | 24.17 | 0 | 11.48 | 0 | 20.69 | 14.81 | 0 | 0 |
| Doubao-pro-4k | 1.56 | 0 | 7.14 | 0.07 | 0 | 0 | 0 | 0 | 1.14 | 0 | 0 | 0 | 0 | 0 | 7.41 | 0.07 | 0 | 0 |
| Doubao-1.5-pro | 0 | 0 | 0 | 0 | 0 | 0 | 0 | 0 | 1.80 | 0 | 0 | 0 | 3.33 | 0 | 0 | 0 | 9.38 | 0 |
| qwen-plus-latest | 0.57 | 0 | 29.95 | 14.29 | 8.83 | 0 | 25.93 | 11.11 | 18.04 | 6.25 | 33.20 | 0 | 30.10 | 20.00 | 17.39 | 11.11 | 21.88 | 12.50 |
| Qwen2.5-max | 53.91 | 18.75 | 31.88 | 14.29 | 11.68 | 0 | 26.85 | 11.11 | 27.71 | 6.25 | 17.22 | 0 | 5.56 | 0 | 16.62 | 11.11 | 0 | 0 |
| DeepSeek-Coder-V2-Lite-Instruct-16B | 0 | 0 | 0 | 0 | 5.56 | 0 | 11.11 | 11.00 | 0 | 0 | 0 | 0 | 0 | 0 | 0 | 0 | 10.94 | 0 |
| DeepSeek-R1 | 36.23 | 6.25 | 29.75 | 14.29 | 32.80 | 11.11 | 28.24 | 11.11 | 35.24 | 20.83 | 32.70 | 0 | 8.99 | 0 | 24.20 | 18.52 | 12.86 | 0 |
| Llama3.1-70B | 41.07 | 18.75 | 25.24 | 14.29 | 14.98 | 0 | 15.67 | 11.11 | 25.13 | 4.17 | 5.11 | 0 | 14.26 | 0 | 21.49 | 11.11 | 0 | 0 |
| Qwen3-8B | 0 | 0 | 2.38 | 0 | 0 | 0 | 11.11 | 11.00 | 2.72 | 0 | 0 | 0 | 0 | 0 | 0 | 0 | 0 | 0 |

Table 12: Detailed results of CORECODEBENCH-*difficult*.

| model/repo | transformers | | langchain | | langchain.core | | datachain | | open-iris | | UniRef | | haystack | | d3rlpy | | inference | | rdt | | cloudnetpy | | skfolio | | finam | |
|---|---|---|---|---|---|---|---|---|---|---|---|---|---|---|---|---|---|---|---|---|---|---|---|---|---|---|
| | AC Rate | AC @1 | AC Rate | AC @1 | AC Rate | AC @1 | AC Rate | AC @1 | AC Rate | AC @1 | AC Rate | AC @1 | AC Rate | AC @1 | AC Rate | AC @1 | AC Rate | AC @1 | AC Rate | AC @1 | AC Rate | AC @1 | AC Rate | AC @1 | AC Rate | AC @1 |
| GPT-4o | 15.85 | 2.86 | 6.25 | 0 | 0 | 0 | 14.95 | 0 | 0 | 0 | 0 | 0 | 8.51 | 0 | 11.43 | 0 | 17.73 | 0 | 66.67 | 0 | 50.00 | 50.00 | 43.39 | 0 | 0 | 0 |
| GPT-4.1 | 18.88 | 2.86 | 2.08 | 0 | 37.50 | 0 | 24.26 | 0 | 0 | 0 | 0 | 0 | 3.58 | 0 | 0 | 0 | 0 | 0 | 0 | 0 | 0 | 0 | 41.44 | 0 | 0 | 0 |
| o4-mini | 0.71 | 0 | 7.85 | 0 | 0 | 0 | 41.30 | 0 | 0 | 0 | 0 | 0 | 7.48 | 0 | 10.00 | 0 | 15.71 | 0 | 66.67 | 0 | 0 | 0 | 70.34 | 25.00 | 0 | 0 |
| Claude-3.7-Sonnet | 46.58 | 28.57 | 31.57 | 0 | 12.50 | 0 | 4.90 | 0 | 0 | 0 | 33.33 | 0 | 12.75 | 0 | 11.43 | 0 | 13.45 | 0 | 66.67 | 0 | 0 | 0 | 52.34 | 0 | 0 | 0 |
| Gemini-2.5-Pro-Preview | 12.69 | 5.71 | 57.37 | 25.00 | 0 | 0 | 4.90 | 0 | 0 | 0 | 33.33 | 0 | 9.21 | 4.00 | 12.78 | 0 | 30.04 | 0 | 0 | 0 | 0 | 0 | 23.83 | 0 | 0 | 0 |
| Grok-3 | 0 | 0 | 37.02 | 0 | 12.50 | 0 | 38.85 | 0 | 0 | 0 | 0 | 0 | 26.28 | 12.00 | 0 | 0 | 17.25 | 0 | 100.00 | 100.00 | 50.00 | 50.00 | 52.73 | 25.00 | 0 | 0 |
| Doubao-pro-4k | 0 | 0 | 0 | 0 | 0 | 0 | 0 | 0 | 0 | 0 | 0 | 0 | 1.37 | 0 | 0 | 0 | 0 | 0 | 0 | 0 | 0 | 0 | 1.92 | 0 | 0 | 0 |
| Doubao-1.5-pro | 0 | 0 | 0 | 0 | 0 | 0 | 0 | 0 | 0 | 0 | 0 | 0 | 2.11 | 0 | 0 | 0 | 0 | 0 | 0 | 0 | 0 | 0 | 1.92 | 0 | 0 | 0 |
| Qwen2.5-max | 0 | 0 | 0 | 0 | 0 | 0 | 0 | 0 | 0 | 0 | 0 | 0 | 0 | 0 | 0 | 0 | 0 | 0 | 100.00 | 100.00 | 0 | 0 | 0 | 0 | 0 | 0 |
| DeepSeek-R1 | 25.71 | 11.43 | 67.95 | 0 | 0 | 0 | 0.37 | 0 | 0 | 0 | 0 | 0 | 11.01 | 0 | 12.86 | 0 | 12.15 | 0 | 100.00 | 100.00 | 0 | 0 | 51.98 | 0 | 0 | 0 |
| Llama3.1-70B | 16.52 | 0 | 12.02 | 0 | 0 | 0 | 0.74 | 0 | 0 | 0 | 0 | 0 | 9.39 | 0 | 11.43 | 0 | 11.54 | 0 | 66.67 | 0 | 0 | 0 | 44.29 | 0 | 0 | 0 |