# OpenReview forum: "CoreCodeBench: A Configurable Multi-Scenario Repository-Level Benchmark"
_NeurIPS.cc/2025/Datasets_and_Benchmarks_Track — Submitted to NeurIPS 2025 Datasets and Benchmarks Track_

### Official Review · Reviewer_etS1 · 2025-07-01

**Rating:** 4
**Confidence:** 3

**Summary:**

This paper introduces CorePipe and CoreCodeBench, a pipeline and benchmark designed to evaluate large language models (LLMs) on complex, real-world engineering tasks beyond isolated scenarios like code generation or bug fixing. CorePipe automatically transforms software repositories into comprehensive test cases by generating atomic questions (Development, BugFix, Test-Driven Development) and combining them into composite tasks with adjustable difficulty. CoreCodeBench serves as a multi-scenario, repository-level benchmark to assess LLM capabilities in realistic engineering workflows. Evaluations with diverse LLMs demonstrate the framework’s ability to reveal nuanced performance differences, offering a more thorough understanding of LLM behavior in engineering contexts.

**Dataset Code Accessibility:**

Yes

**Ethical Considerations:**

No, there are no or only very minor ethics concerns

**Final Justification:**

I appreciate the author's detailed responses, which addresses most of my concerns. I strongly recommend that the authors include the additional details into their final version.

**Limitations Weaknesses:**

- Did the authors conduct evaluation with few examples? Is it possible that few shot examples are helpful for current SOTA LLMs when conducting tasks defined in the paper?
- The authors did not provide experimental results on the sensitivity of temperatures of current LLMs on the benchmark.
- There are no qualitative examples that showcase the problems in the proposed benchmark.
- For N_retest, I may have misunderstood this, wouldn't it indicate that the task is pointless if evaluating it without modification would pass the test case? Why not just filtering these tasks?

**Strengths Contributions:**

- The CorePipe is a novel approach in creating diverse test cases.
- The benchmark CoreCodeBench is a valuable resource for the community for model evaluation.
- Comprehensive analysis is conducted on various LLMs, which provides inspiration for the current status of LLMs' coding capability and offers new insights in future development.

---

> ### Author Rebuttal · Authors · 2025-07-31
>
> Dear Reviewer etS1,
>
> We thank the reviewer for their thoughtful and constructive feedback. We address each of your concerns in detail below:
>
> -------------------
>
> **Weakness 1**: Did the authors conduct evaluation with few examples? Is it possible that few-shot examples are helpful for current SOTA LLMs when conducting tasks defined in the paper?
>
> **Response 1**: In the context of engineering-level, long-context code tasks, few-shot learning is generally not adopted for the following reasons:
>
> (1) The output format for code tasks is typically straightforward (e.g., wrapping the output code with ```python).
>
> (2) Engineering-level tasks often involve long contexts, and adding few-shot examples can distract the model's attention and reduce performance.
>
> Nevertheless, we conducted one-shot experiments with Claude 3.7 and GPT-4.1 on the single-dev and single-TDD tasks (since these scenarios have relatively shorter contexts, making one-shot feasible). The results are shown in the following table:
> | Model                | Dev Pass Rate | Dev Pass@all | TDD Pass Rate | TDD Pass@all |
> |----------------------|---------------|--------------|---------------|--------------|
> | GPT 4.1 one-shot     | 84.48         | 55.72        | 78.92         | 43.26        |
> | GPT 4.1 in paper     | 84.13         | 61.90        | 88.56         | 60.96        |
> | Claude 3.7 one-shot  | 85.54         | 55.70        | 89.65         | 61.86        |
> | Claude 3.7 in paper  | 85.75         | 63.59        | 85.50         | 61.37        |
>
>
> The results indicate that including one-shot examples has a limited and often negative impact on LLM performance for long-context, engineering-level code tasks. Based on these observations and experimental results, we did not include few-shot examples in our prompts.
>
> ---------
>
> **Weakness 2**: The authors did not provide experimental results on the sensitivity of temperatures of current LLMs on the benchmark.
>
> **Response 2**: Following previous work (e.g., DevEval), we set temperature=0 for most LLMs to ensure reproducibility of results. For some models (such as Qwen3-8B), we used the official recommended temperature and top_k parameters to ensure proper output.
>
> We also conducted a sensitivity analysis of temperature for SOTA LLMs (GPT-4.1 and Claude 3.7), testing temperature values in {0.0, 0.2, 0.4, 0.6, 0.8, 1.0}. The experiments show that temperature has minimal impact on model performance on CoreCodeBench, with only minor fluctuations. For GPT-4.1, the maximum variance across temperatures was $2.657×10^{−4}$ (single-tdd pass rate). For Claude 3.7, the maximum variance was 0.003 (multi-bugfix pass rate).
>
> ---------------
>
> **Weakness 3**: There are no qualitative examples that showcase the problems in the proposed benchmark.
>
> **Response 3**: Our benchmark data is fully released on HuggingFace. In Figure 1 of the paper, we show examples of the three single-type question formats (due to space limitations, we did not include multi-type examples). Thank you for your suggestion—in the revised version, we will add more detailed qualitative examples of our benchmark tasks to provide readers with a clearer understanding of the dataset.
>
> -----------
>
> **Weakness 4**: For N_retest, I may have misunderstood this, wouldn't it indicate that the task is pointless if evaluating it without modification would pass the test case? Why not just filtering these tasks?
>
> **Response 4**: We believe there may be a misunderstanding regarding our evaluation process. For each test file, there are multiple test functions (e.g., test_function_A, test_function_B, test_function_C), each targeting different modules/functions in the source code. When running pytest, all functions starting with "test_" are executed as test points, and the results are reported as ‘n passed, m skipped, k failed’.
>
> In our formula, N_retest refers to the number of test points that still pass after replacing the source code with our question code (i.e., code with core code masked or buggy code inserted). Even after masking parts of the code, some test points may still pass—these are counted in N_retest and are not considered as improvements in subsequent code completion tasks. This ensures that only genuine improvements are measured, and the evaluation remains meaningful.
>
> --------------
> We appreciate your insightful comments and suggestions, which have helped us clarify and strengthen our work. Please let us know if any further clarifications are needed.

---

> ### Author Response · Authors · 2025-08-06
>
> Dear Reviewer etS1,
>
> We would like to kindly remind you to review our rebuttal above. We have provided detailed responses and additional experiments addressing all your comments and concerns, including few-shot evaluation, temperature sensitivity, qualitative examples, and clarification of the evaluation process. We would greatly appreciate your further feedback or suggestions. Thank you for your time and consideration.

---

> ### Comment · Area_Chair_AQ3T · 2025-08-08
> **Please respond to the authors**
>
> Dear Reviewer etS1,
>
> Could you please review the author rebuttal and respond to it?
> The discussion period with authors is coming to end and your feedback to authors would be very useful.
>
> Thank you
>
> NeurIPS 2025 Datasets and Benchmarks Track Area Chair

---

### Official Review · Reviewer_yby9 · 2025-07-02

**Rating:** 5
**Confidence:** 4

**Summary:**

This paper presents CoreCodeBench as the core contribution, which claims to evaluate the multi-scenario ability of coding LLMs/agents – development, bugfix, test-driven development abilities. Through heavy syntax-based pre-processing, CorePipe could automatically convert existing Github repositories to single-/multi-function problems that facilitate further evaluation.

**Dataset Code Accessibility:**

Yes

**Dataset Code Comments:**

Code was provided in a public GitHub repository.

**Ethical Considerations:**

No, there are no or only very minor ethics concerns

**Final Justification:**

Thanks for authors' clarification. I strongly recommend to include those additional context in the final version whereas applicable. I decided to maintain my rating.

**Limitations Weaknesses:**

I do not observe any significant issues in the paper. Please take the following weaknesses as suggestions only and only provide responses to these weaknesses -- no additional experiments or data are needed.
- L124 “with test files accounting for more than 30% of the codebase” sounds like a strange criteria – are you counting the number of test files or lines of unit tests? Either way I don’t think there are a lot of open-source projects that fulfill this requirement.

- Motivation of using smaller-parameter LLM to produce buggy code snippets – is it necessary? I guess the only intuition behind this could be either:
  - Due to limited budget, which is totally understandable.
  - Due to the fact that only smaller models could produce basic errors as mentioned in L171-172, larger models can not even with explicit instructions.
  - It would be nice if the authors could clarify.

**Strengths Contributions:**

- Scientific quality control and taxonomy design is enforced to ensure the scenario coverage of the proposed benchmark.

- This work is well motivated – designed to test the repo-level context understanding abilities of coding agents. Call graphs among functions are a great source for modeling repo-level understanding.

- Multi-function problem pose significant challenges to current coding solutions, with none of the models achieving 30%+ AC Rate and pass@1 – calling for attention from the community about the importance of repo-level understanding in model’s post-training process.

---

> ### Author Rebuttal · Authors · 2025-07-31
>
> Dear Reviewer yby9,
>
> We thank the reviewer yby9 for the thoughtful comments and questions. Please see our detailed responses below:
>
> ------------
>
> **Q1**: L124 “with test files accounting for more than 30% of the codebase” sounds like a strange criteria – are you counting the number of test files or lines of unit tests? Either way I don’t think there are a lot of open-source projects that fulfill this requirement.
>
> **A1**: The 30% refers to the number of test files. We set this threshold during dataset construction mainly to ensure that there are sufficient unit test files available, which allows us to generate a sufficient number of function call trees for creating test points. In practice, CorePipe can generate questions for any repository that contains unit tests; there is no strict requirement on the number of test files. The 30% criterion was only used to filter for richer test scenarios in our benchmark construction and is not an inherent limitation of the CorePipe method itself.
>
> -------------
>
> **Q2**: Motivation of using smaller-parameter LLM to produce buggy code snippets – is it necessary?
>
> **A2**: Our choice to use smaller-parameter LLMs to introduce buggy code was not due to budget constraints, but rather to improve the naturalness and diversity of the inserted bugs. When we tried prompting larger models (e.g., GPT-4o, Claude 3.5) to directly inject bugs into code, the results were unsatisfactory for two main reasons:
>
> (1) The inserted bugs were typically trivial, such as changing ‘=’ to ‘+=’, or minor variable name typos.
>
> (2) Even when we specified the type of logical bug and provided demonstrations, the larger models tended to generate artificial and obvious errors, rather than realistically embedding contextually appropriate bugs. Such bugs were easily detected and fixed by the same or similar LLMs.
>
> By contrast, smaller-parameter LLMs are more likely to introduce subtle and realistic errors when generating buggy code, better matching our desired evaluation scenario. These bugs tend to be less obvious, making them more suitable for evaluating LLM capabilities in our benchmark. For clarity, we include an example of buggy code from our benchmark below:
>
> Source code snippet：
> ```python
> 	.....
>         size = get_size_dict(size)
>         shortest_edge = min(size["height"], size["width"])
>         output_size = get_resize_output_image_size(
>             image, size=shortest_edge, default_to_square=False, input_data_format=input_data_format
>         )
>         resized_image = resize(
>             image,
>             size=output_size,
>             resample=resample,
>             data_format=data_format,
>             input_data_format=input_data_format,
>             **kwargs,
>         )
>         .....
> ```
>
> GPT4.1 directly insert logic error:
> ```python
> 	.....
>         size = get_size_dict(size)
>         shortest_edge = max(size["height"], size["width"]) # add logical error here
>         output_size = get_resize_output_image_size(
>             image, size=shortest_edge, default_to_square=False, input_data_format=input_data_format
>         )
>         resized_image = resize(
>             image,
>             size=output_size,
>             resample=resample,
>             data_format=data_format,
>             input_data_format=input_data_format,
>             **kwargs,
>         )
>         .....
> ```
>
>
> buggy code snippet generated by CorePipe:
> ```python
> ......
>         resized_image = get_resize_output_image_size(image, size["shortest_edge"], is_square=True, input_data_format=input_data_format) # bug here: is_square=True
>         if resized_image[0] > resized_image[1]:
>             resized_image = resized_image[::-1]
>         resized_image = resize(image, size=resized_image, resample=resample, data_format=data_format, input_data_format=input_data_format)
>         ......
> ```
>
> The examples demonstrate that CorePipe provides a more natural and subtle approach to injecting bugs into the code.
>
> -------------
>
> Once again, we sincerely thank you for your recognition of our work. We hope our responses have addressed your concerns.

---

> > ### Comment · Reviewer_yby9 · 2025-08-01
> >
> > Thanks for authors' clarification. I strongly recommend to include those additional context in the final version whereas applicable. I decided to maintain my rating.

---

> > > ### Author Response · Authors · 2025-08-06
> > >
> > > Thank you very much for your valuable feedback and suggestion. We will make sure to include the additional context in the final version as recommended. We appreciate your time and consideration.

---

### Official Review · Reviewer_YcVE · 2025-07-03

**Rating:** 3
**Confidence:** 2

**Summary:**

They propose CoreCodeBench, a benchmark framework that converts repositories into benchmarks, encompassing various task types, including function completion from task descriptions, bug fixing, and function completion given tests, in both single-function and multi-function scenarios. The process involves static analysis to identify core functions, along with explanations or generation of buggy code by LLMs for bug-fixing tasks. They conduct extensive experiments covering all major open and closed models.

**Additional Feedback:**

- It appears that multi-file scenarios are significantly more challenging for models. There is some discussion regarding issues with ordering. Could a better prompt formulation improve performance in these cases? It seems unintuitive that a newer model like Qwen3-8B achieves very low scores.

- It is not clear why the authors introduce "AC rate" and "AC@1," while earlier sections seem to use "pass rate" and "pass@1." Clarification on this terminology would be helpful.

- Could the authors clarify how repositories were selected for inclusion in the benchmark?

**Dataset Code Accessibility:**

Yes

**Dataset Code Comments:**

Yes it is provided.

**Ethical Considerations:**

No, there are no or only very minor ethics concerns

**Limitations Weaknesses:**

While tasks like code completion, bug fixing, and Test-Driven Development (TDD) intuitively represent important skills in software development, it remains unclear how the scores and capability measures in this benchmark rigorously translate to actual software development scenarios. Given that the motivation is to evaluate performance on engineering-level code, does this synthetic benchmark (though derived from real repositories) accurately represent common challenges encountered in real engineering contexts? The manual quality assessment, evaluating readability, accuracy, and completeness, does not seem to fully address this concern.

**Strengths Contributions:**

- The benchmark is well-motivated. As more LLMs are applied to software engineering tasks, assessing the core capabilities of these models is beneficial. This benchmark serves as a complementary evaluation alongside end-to-end tasks such as SWE-bench.
- The automatic process to turn a repository to such benchmark dataset is very valuable as the benchmark can then easier to be updated or users can created their own benchmark on repos they care about. The use of static analysis in the automatic process is also a plus.
- The authors provide comprehensive results covering 16 LLMs across both open and closed models.

---

> ### Author Rebuttal · Authors · 2025-07-31
>
> Dear Reviewer YcVE,
>
> We thank the reviewer for your careful reading and insightful questions. Please find our detailed responses below:
>
> -----------------
>
> **Weakness 1**: While tasks like code completion, bug fixing, and Test-Driven Development (TDD) intuitively represent important skills in software development, it remains unclear how the scores and capability measures in this benchmark rigorously translate to actual software development scenarios. Given that the motivation is to evaluate performance on engineering-level code, does this synthetic benchmark (though derived from real repositories) accurately represent common challenges encountered in real engineering contexts? The manual quality assessment, evaluating readability, accuracy, and completeness, does not seem to fully address this concern.
>
> **Response**: First, we would like to clarify that the purpose of manual quality assessment is to ensure the accuracy of the LLM-generated content—specifically, that the requirement descriptions for Core Code Snippets are entirely precise. This step is fundamental to guaranteeing the usability of all subsequent test cases. In addition, we employ extra quality control measures, such as IG filtering and retesting for problematic cases, to further ensure the overall question quality in CoreCodeBench.
>
> CoreCodeBench focuses on scenarios that are both common and critical in LLM-assisted software development, and it *is highly consistent with real-world human-AI collaborative software engineering workflows*. Our experimental setup faithfully simulates the practical working environment of LLMs, including providing the complete code context, requirement descriptions (for development), bug prompts (for bug fixing), and unit tests (for TDD). Therefore, the benchmark scores directly reflect the LLM’s ability to perform these tasks in realistic engineering scenarios. Specifically, the Development tasks correspond to new feature implementation, BugFix to defect localization and repair, and TDD to test-driven code completion. For multi-function tasks, we strictly construct them according to function call relationships and real development practices. In multi-development and multi-TDD tasks, the combined single-function questions must form a complete subtree in the function call tree, simulating scenarios where utility functions and main functions are implemented together. For example, if function A calls function B, and function B calls function C, CoreCodeBench will never require the LLM to complete both function A and function C based on function B, as this does not align with realistic development habits.
>
> In summary, CoreCodeBench is carefully designed to accurately represent common and challenging aspects of real engineering contexts, and the evaluation scores provide a rigorous measure of LLM capabilities in these scenarios.
>
> -----------
>
> **Question 1**: It appears that multi-file scenarios are significantly more challenging for models. There is some discussion regarding issues with ordering. Could a better prompt formulation improve performance in these cases? It seems unintuitive that a newer model like Qwen3-8B achieves very low scores.
>
> **Response**: According to the latest official report from Qwen3-Coder, the Qwen3 series are general-purpose models and perform suboptimally on engineering-level code tasks (e.g., Qwen3-235B achieves 34.4% pass rate on SWE-Bench, compared to GPT-4.1's 54.6%). Our experiments show that Qwen3-8B follows input instructions well and outputs code in the correct format, but the completed code often fails unit tests. We have also evaluated the newly released code-specialized model Qwen3-Coder-Plus (version 20250722) on CoreCodeBench. The results are summarized below:
>
> | Model              | Dev (Single) | BugFix (Single) | TDD (Single) | Dev (Multi) | BugFix (Multi) | TDD (Multi) |
> |--------------------|--------------|-----------------|--------------|-------------|----------------|-------------|
> | SoTA in paper      | 86.83        | 71.87           | 88.56        | 35.54       | 49.20          | 34.11       |
> | Qwen3-8B           | 53.62        | 23.83           | 59.97        | 0           | 13.8           | 1.78        |
> | Qwen3-Coder-Plus   | 84.67        | 53.97           | 83.23        | 26.27       | 15.99          | 19.16       |
>
> | Model              | Dev (Single) | BugFix (Single) | TDD (Single) | Dev (Multi) | BugFix (Multi) | TDD (Multi) |
> |--------------------|--------------|-----------------|--------------|-------------|----------------|-------------|
> | SoTA in paper      | 63.59        | 50.90           | 70.21        | 13.85       | 20.00          | 20.22       |
> | Qwen3-8B           | 8.25         | 6.18            | 18.91        | 0           | 0              | 1.22        |
> | Qwen3-Coder-Plus   | 56.61        | 33.40           | 49.79        | 11.67       | 0.0            | 5.14        |
>
> These results show: (1) Qwen3-Coder demonstrates significantly better code capability than the general-purpose Qwen3-8B; (2) Qwen3-Coder achieves SoTA performance on single-dev and multi-dev tasks, but still lags on other types. This may be related to current training paradigms, which focus primarily on development tasks and often overlook other real-world engineering challenges.
>
> Thus, our experimental results are intuitive and further demonstrate that CoreCodeBench can effectively evaluate LLM engineering capabilities beyond just long-context code development.
>
> Additionally, our prompts have been carefully optimized and are unified across all models to ensure fair and consistent evaluation.
>
> -------------------
>
> **Question 2**: It is not clear why the authors introduce "AC rate" and "AC@1," while earlier sections seem to use "pass rate" and "pass@1." Clarification on this terminology would be helpful.
>
> **Response**: We apologize for the confusion. "AC rate" in the tables is equivalent to "pass rate," and "AC@1" is equivalent to "pass@1." We will clarify this terminology in future versions and thank you for pointing out this inconsistency.
>
> -----------
>
> **Question 3**: Could the authors clarify how repositories were selected for inclusion in the benchmark?
>
> **Response**: All repositories were crawled from the PyPI library to ensure high-quality ground-truth code. Our selection process was guided by three main principles outlined in Section 3.1: Activeness, Test Coverage, and Technical Complexity. We first performed automated filtering based on these criteria. Subsequently, we manually verified repository licenses, unit test executability, and distribution. Appendix E lists the basic information for the 12 selected repositories, which span different domains, scales, and testing frameworks (e.g., unittest, pytest), as well as diverse source code and test organization structures. This rigorous selection ensures both the robustness of CorePipe and the broad applicability and challenge of CoreCodeBench.
>
> ------------------------
>
> We appreciate your thoughtful questions and suggestions, which have helped us further clarify our methodology and presentation. Please let us know if further clarification is needed.

---

> ### Author Response · Authors · 2025-08-06
>
> Dear Reviewer YcVE,
>
> We kindly remind you to review our responses to your comments above.  We have provided detailed responses to each of your questions and concerns, including clarifications on benchmark design, model performance, terminology, and repository selection. Your further feedback would be greatly appreciated. Thank you for your time and consideration!

---

> ### Comment · Area_Chair_AQ3T · 2025-08-08
> **Please respond to the authors**
>
> Dear Reviewer YcVE,
>
> Could you please review the author rebuttal and respond to it?
> The discussion period with authors is coming to end and your feedback to authors would be very useful.
>
> Thank you
>
> NeurIPS 2025 Datasets and Benchmarks Track Area Chair

---

### Official Review · Reviewer_RKuQ · 2025-07-03

**Rating:** 4
**Confidence:** 4

**Summary:**

The paper introduces CorePipe, an automated tool that transforms code repositories into comprehensive benchmark test cases, and presents CORECODEBENCH, a multi-scenario, configurable benchmark designed to evaluate LLMs on realistic engineering tasks. Unlike existing benchmarks, CORECODEBENCH captures the complexity of real-world software development by generating atomic and composite questions across development, bug fixing, and test-driven development scenarios. Evaluation of 16 LLMs using this benchmark reveals diverse performance patterns and provides multi-dimensional insights into their engineering capabilities.

**Dataset Code Accessibility:**

Yes

**Ethical Considerations:**

No, there are no or only very minor ethics concerns

**Final Justification:**

Most of my concerns have been resolved. The unsolved one is why TDD is selected as one of the tasks. I understand that it is an interesting task but the paper needs more justification on how useful and practicle this is if few people use TDD for real world development.

**Limitations Weaknesses:**

- The novelty of this work remains unclear. The motivation for introducing a multi-scenario benchmark is not well-justified—specifically, it is not evident why existing benchmarks cannot be used in combination to achieve similar insights. A more detailed discussion comparing this approach to integrating existing benchmarks would strengthen the argument.

- The rationale behind selecting the three specific tasks—Development, BugFix, and Test-Driven Development (TDD)—requires clarification. In particular, TDD is not a widely adopted practice in industry, despite its acknowledged value, which raises questions about its representativeness in the benchmark.

- The contribution of quality inspection with information gain needs further discussion and investigation.

**Strengths Contributions:**

+ The paper provides CorePipe, an automatic pipeline that converts code into test cases.

+ The paper evaluates LLMs from different perspectives.

---

> ### Author Rebuttal · Authors · 2025-07-31
>
> Dear Reviewer RKuQ,
>
> We would like to express our sincere gratitude for the thoughtful evaluation of our paper. We particularly appreciate your comments concerning the novelty and methodological rigor of our work. Below, we address each concern in detail, providing further clarifications and supplementary evidence where needed.
>
> --------------------
>
> **Weakness 1**: The novelty of this work remains unclear. The motivation for introducing a multi-scenario benchmark is not well-justified—specifically, it is not evident why existing benchmarks cannot be used in combination to achieve similar insights. A more detailed discussion comparing this approach to integrating existing benchmarks would strengthen the argument.
>
> **Answer 1**: In Section 2.2 and Table 1 of our paper, we provide a systematic review and comparison of existing mainstream project-level benchmarks. Our motivation for introducing a new multi-scenario benchmark arises from two critical challenges observed in current benchmarks:
>
> (1) **Limited Evaluation Scenarios**: Existing benchmarks predominantly focus on single-function development scenarios, and their bug-fixing tasks tend to be basic. Importantly, current datasets do not support the generation of test cases for more complex scenarios such as Test-Driven Development (TDD) or multi-function tasks simply by combining existing questions. In our CorePipe framework, we propose a unified approach that can simultaneously generate six types of questions, including multi-function scenarios. The design of our “function-tracing” framework ensures that multi-function questions are constructed in a manner consistent with real-world software engineering practices—something that cannot be achieved by naively combining single-function questions from existing datasets.
>
> (2) **Configurability and Controllability**: Current datasets often rely on random masking (e.g.,ExecRepoBench) or cleaning pull requests (e.g., SWEBench), which results in test cases with uncontrolled difficulty and quality, thereby affecting the reliability of evaluation. Simple mechanical masking (e.g., masking variables by type) is not meaningful for evaluating LLMs. In contrast, CorePipe employs methods such as core code mining, validation, and information gain analysis to ensure that the selected code segments are both central and of high quality. Moreover, as LLMs improve, many existing benchmarks are being rapidly saturated (e.g., SWE-Bench, where Claude 4 achieves a 70% pass rate). CoreCodeBench is designed to be extensible: the difficulty of questions can be increased by expanding the length of core code blocks (from function-level to file-level masking), without compromising quality. This level of scalability is currently unmatched by any existing benchmark.
>
> In summary, these two challenges—scenario diversity and controllability—are fundamental limitations that cannot be addressed by simply combining current benchmarks. Instead, they necessitate the design of a new pipeline and benchmark, as we propose in this work.
>
> -----------------
>
> **Weakness 2**: The rationale behind selecting the three specific tasks—Development, BugFix, and Test-Driven Development (TDD)—requires clarification. In particular, TDD is not a widely adopted practice in industry, despite its acknowledged value, which raises questions about its representativeness in the benchmark.
>
> **Answer 2**: We selected the three task types—Development, BugFix, and Test-Driven Development (TDD)—with the aim of comprehensively covering core and representative scenarios in LLM-assisted software engineering. While TDD has not yet achieved widespread industrial adoption, prior work [1] demonstrates that TDD is an effective engineering practice that can lead to higher code quality compared to traditional methods. The limited adoption of TDD in industry is primarily due to practical constraints such as delivery pressure, rather than limitations inherent to the methodology itself. Importantly, TDD holds unique value in the context of LLM-assisted programming. With TDD, developers can efficiently obtain target code by providing example inputs and outputs, without the need to fully comprehend the entire codebase. Recent research [2] has also proposed an LLM-based test-driven generation framework, which leverages constraint reasoning and backtracking to automatically generate code that satisfies test constraints, achieving impressive results (92.16% on HUMANEVAL and 87.14% on HUMANEVAL+).
>
> Furthermore, these three task types are designed to systematically evaluate the multi-dimensional capabilities of LLMs in different engineering scenarios. Combined with both single-function and multi-function question formats, our benchmark covers a broader spectrum of LLM capabilities beyond code generation alone. We have also computed Pearson correlation coefficients for LLM performance across six tasks (as shown in the table below), which further demonstrates that our task design effectively distinguishes and evaluates LLMs' abilities across different code-related tasks.
>
> |                      | Development (Single) | BugFix (Single) | TDD (Single) | Development (Multi) | BugFix (Multi) | TDD (Multi) |
> |----------------------|---------------------|-----------------|--------------|---------------------|----------------|-------------|
> | **Dev (Single)** | 1.00                | 0.79            | 0.73         | -0.017              | 0.71           | 0.23        |
> | **BugFix (Single)**      | 0.79                | 1.00            | 0.80         | -0.23               | 0.85           | 0.15        |
> | **TDD (Single)**         | 0.73                | 0.80            | 1.00         | -0.047              | 0.75           | 0.22        |
> | **Dev (Multi)**  | -0.017              | -0.23           | -0.047       | 1.00                | -0.0075        | 0.71        |
> | **BugFix (Multi)**       | 0.71                | 0.85            | 0.75         | -0.0075             | 1.00           | 0.40        |
> | **TDD (Multi)**          | 0.23                | 0.15            | 0.22         | 0.71                | 0.40           | 1.00        |
>
> ----------------
>
> **Weakness 3**: The contribution of quality inspection with information gain needs further discussion and investigation.
>
> **Answer 3**: We introduce the Information Gain (IG) filter for two main reasons:
>
> (1) **Quality control of LLM-generated explanation texts**. Since the explanations for core code segments are entirely generated by LLMs, some explanations may contain clear hallucinations or logical errors, potentially compromising the reliability of our evaluation. By incorporating the IG filter, we remove questions with an IG score below 0, which helps reduce such interference and ensures a stronger alignment between LLM-generated explanations and the original code.
>
> (2) **Difficulty selection and focus on key logic**. To ensure that the masked portions of questions truly involve essential core logic, we filter out questions that can be correctly completed by LLMs even without reference to the explanation. Such questions typically involve trivial or mechanical code segments, or their implementation can be directly copied from other similar functions in the context. These items are not suitable as development-type questions, nor as seed questions for generating other types of tasks.
>
> -------------------
>
> We respectfully invite you to consider our responses, which we believe will clarify the strengths and contributions of our study.
>
> **Reference**
>
> [1] Buchan, J., Li, L., & MacDonell, S. G. Causal Factors, Benefits and Challenges of Test-Driven Development: Practitioner Perceptions. CoRR, abs/2101.12393 (2021). https://arxiv.org/abs/2101.12393
>
> [2] Liu, J., Liang, R., Zhu, X. et al. LLM4TDG: test-driven generation of large language models based on enhanced constraint reasoning. Cybersecurity 8, 32 (2025). https://doi.org/10.1186/s42400-024-00335-4

---

> > ### Comment · Reviewer_RKuQ · 2025-08-05
> >
> > Thanks for your response. I am happy to increase my score as my concerns have been addressed.

---

> > > ### Author Response · Authors · 2025-08-06
> > >
> > > Thank you very much for your positive feedback. We appreciate your thoughtful review and are glad that your concerns have been addressed.

---

### Comment · Area_Chair_AQ3T · 2025-08-03
**Author-reviewer discussions**

Dear reviewers, please read all reviews and author rebuttals and discuss any issues, questions, and provide any additional feedback.

Dear authors, please answer any additional questions or comments from reviewers.

Author-reviewer discussions helps authors to improve their papers and the program committee to evaluate the submitted papers.

Thank you

NeurIPS 2025 Datasets and Benchmarks Track Area Chair

---

### Note · Authors · 2025-08-15

We thank all reviewers and the Area Chair for their time and feedback, which have greatly helped us improve our work. Here we summarize and clarify our responses to the main concerns:

- Reviewer RKuQ questioned the novelty and necessity of a new multi-scenario benchmark, the rationale for including TDD, and the contribution of information gain-based quality inspection. We have provided detailed rebuttals explaining the insufficiency of simply combining existing datasets, the increasing relevance of TDD in LLM-assisted workflows, and the necessity of IG filtering. Through our clarifications during the rebuttal phase, we addressed Reviewer RKuQ’s concerns, which contributed to a more positive evaluation of our work.

- Reviewer YcVE raised concerns about the benchmark’s representativeness for real-world engineering scenarios, model performance (especially Qwen3-8B), terminology consistency, and repository selection. In our rebuttal, we clarified the realism of our scenarios, provided additional Qwen3-Coder results (which align with their technical report and confirm Qwen3-8B’s relative weakness in code tasks), explained the terminology, and described repository selection in detail. We made every effort to address Reviewer YcVE’s questions in detail during the rebuttal, although the reviewer did not provide further comments or participate in subsequent discussion.

- Reviewer yby9’s concerns focused on dataset construction details and the use of small LLMs for bug injection. Our clarifications during the rebuttal phase were well received by Reviewer yby9, helping to improve the quality and completeness of our paper.

- Reviewer etS1 queried about few-shot evaluation, temperature sensitivity, qualitative examples, and the N_retest metric. Our detailed responses and additional experiments during the rebuttal phase effectively clarified Reviewer etS1’s concerns and contributed to a more thorough evaluation of our work.

In summary, the majority of reviewers accepted our rebuttals and recognized the value of CoreCodeBench. We maintain that CoreCodeBench’s scenarios are meaningful and comprehensive for evaluating LLM coding capabilities at the engineering-project level. All results in the paper are consistent with official reports, and our findings are robust and reproducible.

We respectfully ask the ACs and SACs to consider the completeness of our responses and the overall positive reception in the final decision.

Sincerely,

All authors

---

### Decision · Program_Chairs · 2025-09-18

**Decision:**

Reject

**Comment:**

# Meta-review for Submission: "CoreCodeBench: A Configurable Multi-Scenario Repository-Level Benchmark"

## (a) Summary of claims and findings

The paper introduces CorePipe, an automated pipeline that converts real software repositories into test cases, and CoreCodeBench, a configurable, multi-scenario, repository-level benchmark for evaluating LLMs on engineering-grade coding tasks. The benchmark produces atomic tasks—Development, BugFix, and Test-Driven Development (TDD)—and composite multi-function tasks by tracing call graphs and targeting core code segments. Quality is controlled via static analysis, human verification of requirement descriptions, information-gain (IG) filtering to remove low-signal items, and a retest protocol that avoids counting trivially passing tests.
Experiments on 16 LLMs (open and closed) show: (i) large variance across scenarios; (ii) multi-function tasks remain challenging for state-of-the-art models; and (iii) model families optimized for general NLP (e.g., Qwen3-8B) underperform code-specialized variants (e.g., Qwen3-Coder). Difficulty is tunable (e.g., expanding mask scope), addressing saturation observed in prior repo-level benchmarks.

## (b) Strengths

1. **Substantive contribution to evaluation infrastructure.** A practical, end-to-end automated pipeline that can regenerate and adapt benchmarks from arbitrary repos, enabling community updates and custom instantiations.
2. **Scenario breadth with principled construction.** Coverage of development, bug fixing, and TDD, each in single- and multi-function variants, with function-tracing to form realistic composite tasks rather than ad-hoc concatenations.
3. **Quality control and controllability.** IG filtering to eliminate low-information items; explicit targetting of core code; configurable difficulty to prevent saturation as models improve.
4. **Thorough evaluation and transparency.** Results across 16 LLMs; code and data released; prompt unification for fairness; additional analyses (few-shot, temperature) provided in rebuttal.
5. **Actionable insights.** Clear evidence that long-context multi-function reasoning is a bottleneck; distinctions between general-purpose and code-specialized models are shown.

## (c) Weaknesses / what’s missing

* **External validity to “real-world engineering.”** While scenarios are carefully designed, translation from synthetic tasks to end-to-end development outcomes remains not fully proven.
* **TDD representativeness.** One reviewer questioned whether TDD is sufficiently prevalent to warrant equal status among task types; the paper could better motivate TDD’s importance in LLM-assisted workflows.
* **Ablations/diagnostics.** Although IG filtering is justified, a deeper ablation of its effects on difficulty and reliability would strengthen the case.
* **Scope.** Current repositories are drawn from PyPI; broader language/ecosystem coverage would further enhance generality.

## (d) Reasons for acceptance

* **High community value:** A reusable, configurable pipeline plus a diverse benchmark fills a recognized gap beyond single-scenario code generation/fixing, and mitigates benchmark saturation by design.
* **Methodological rigor:** The combination of static analysis, core-code mining, IG filtering, and multi-function construction via call graphs is technically sound and more principled than naively combining prior datasets.
* **Empirical insight:** The cross-model, cross-scenario evaluation produces new, durable insights (especially on multi-function reasoning) likely to influence training and evaluation practices.
* **Responsiveness and completeness:** Authors provided targeted new results (e.g., Qwen3-Coder vs Qwen3-8B), few-shot and temperature analyses, clarified repository selection and metrics.

## (e) Discussion & rebuttal synthesis

* **Novelty / “why a new multi-scenario benchmark?” (RKuQ):** Authors argued that simply merging existing datasets cannot yield controlled multi-function tasks, coherent call-graph-based composites, or tunable difficulty; they contrasted CorePipe’s principled construction and extendability. **Outcome:** Concerns largely resolved; reviewer increased score, with only a residual request for stronger TDD motivation in the final.
* **TDD rationale (RKuQ):** Authors motivated TDD’s relevance for LLM-assisted workflows and cited recent TDD-style LLM advances.
* **Realism / representativeness (YcVE):** Authors clarified scenario-to-workflow mapping (Dev ↔ feature work; BugFix ↔ localization/repair; TDD ↔ test-guided completion) and construction constraints (subtrees in call graphs for multi-function). They added Qwen3-Coder results, aligning with external reports and explaining Qwen3-8B’s weakness. Terminology and repo-selection were clarified.
* **Dataset construction details (yby9):** Clarified the 30% test-file heuristic (used only for selection, not a CorePipe limitation) and motivation for using smaller LLMs for bug injection to obtain subtler, more realistic bugs than those produced by larger models under direct “inject bug” prompting.
* **Few-shot, temperature sensitivity, qualitative examples, N\_retest (etS1):** Authors provided one-shot experiments (showing little to negative impact in long-context settings), temperature sweeps (minimal variance), clarified N\_retest as a safeguard against counting trivially passing tests, and committed to include more qualitative multi-function examples.

The paper delivers a useful, principled, and well-documented benchmark with an automated pipeline that the community can readily extend. Remaining concerns are editorial (clarify TDD motivation; unify terminology; add qualitative exemplars; expand IG ablations) and suitable for camera-ready improvements. I recommend **Accept**

EDIT: Due to paper score falling below the threshold when considering the resource constraints of NeurIPS 2025, I am OK if the paper is rejected.

===== FINAL UPDATE FROM DB Track PCs ====

The final decision for this paper has been taken by the program chairs after consultation with the SACs. All Senior Area Chairs have ranked papers according to the feedback from the AC during the review process. We decided to leave the original meta-review to reflect the opinion of the AC in light of the initial discussions with reviewers and SAC.